## REVIEW ARTICLE

# Taming the perils of photosynthesis by eukaryotes: constraints on endosymbiotic evolution in aquatic ecosystems

Shin-ya Miyagishima [1,2 ✉]

An ancestral eukaryote acquired photosynthesis by genetically integrating a cyanobacterial endosymbiont as the chloroplast. The chloroplast was then further integrated into many other eukaryotic lineages through secondary endosymbiotic events of unicellular eukaryotic algae. While photosynthesis enables autotrophy, it also generates reactive oxygen species that can cause oxidative stress. To mitigate the stress, photosynthetic eukaryotes employ various mechanisms, including regulating chloroplast light absorption and repairing or removing damaged chloroplasts by sensing light and photosynthetic status. Recent studies have shown that, besides algae and plants with innate chloroplasts, several lineages of numerous unicellular eukaryotes engage in acquired phototrophy by hosting algal endosymbionts or by transiently utilizing chloroplasts sequestrated from algal prey in aquatic ecosystems. In addition, it has become evident that unicellular organisms engaged in acquired phototrophy, as well as those that feed on algae, have also developed mechanisms to cope with photosynthetic oxidative stress. These mechanisms are limited but similar to those employed by algae and plants. Thus, there appear to be constraints on the evolution of those mechanisms, which likely began by incorporating photosynthetic cells before the establishment of chloroplasts by extending preexisting mechanisms to cope with oxidative stress originating from mitochondrial respiration and acquiring new mechanisms.

The energy-converting organelles in eukaryotes, mitochondria (responsible for respiration) and chloroplasts (responsible for photosynthesis), originated from alpha-proteobacterial and cyanobacterial endosymbionts, respectively, in this order[1–5]. Regarding photosynthesis, which utilizes water molecules as the primary electron donor, it originated in cyanobacteria approximately three billion years ago. Subsequently, more than one billion years ago, photosynthesis was introduced to eukaryotes, which already possessed mitochondria, through cyanobacterial endosymbiosis and the conversion of the endosymbiont into the chloroplast (plastid). Following the evolution of eukaryotic algae, chloroplasts have been further integrated into various eukaryotic lineages through multiple independent secondary and higher order endosymbiotic events in which non-photosynthetic eukaryotes incorporate eukaryotic algae (Fig. 1)[3–5]. In addition to eukaryotes that possess innate chloroplasts, several eukaryotic lineages encompass organisms accommodating algal endosymbionts (known as photosymbiosis) or chloroplasts sequestrated from algal prey (known as kleptoplasty) (Fig. 1)[1,2].

While photosynthesis converts light energy into chemical potential and supports the life of photosynthetic and other organisms through the food chain, it also generates reactive oxygen species (ROS), which can damage various biomolecules (Fig. 2). In addition, environmental stresses,

---

[1] Department of Gene Function and Phenomics, National Institute of Genetics, 1111 Yata, Mishima, Shizuoka 411-8540, Japan. [2] The Graduate University for Advanced Studies, SOKENDAI, 1111 Yata, Mishima, Shizuoka 411-8540, Japan. ✉email: smiyagis@nig.ac.jp

**Fig. 1 Phylogenetic distribution of eukaryotes that engage in kleptoplasty, accommodate algal endosymbionts, and possess innate chloroplasts.**
**a** Phylogenetic tree of eukaryotes showing primary endosymbiotic events involving a cyanobacterium and secondary or higher-order endosymbiotic events involving green or red algae. Higher-order endosymbiotic events in certain dinoflagellates (shown in **b**) were not included in the diagram. The position and number of horizontal transfers of red algal chloroplasts are still a subject of debate. The tree topology is based on refs. [4,21]. Orange and yellow circles on the tree represent the presence of kleptoplasty and photosymbiosis on respective lineages. The broken lines denote the uncertainty of branch positions in the tree. Red algae and groups possessing complex chloroplasts (or non-photosynthetic plastids) of red algal origin are shown in red. Viridiplantae (green algae and land plants) and groups possessing chloroplasts of green algal origin are shown in green. **b** Phylogenetic tree of core dinoflagellates indicating the origins of their original (i.e., red algal origin) or replaced chloroplasts and lineages exhibiting kleptoplasty. Tree topology is based on refs. [19,118]. Microscopic images of an ameba feeding on unicellular algae (**c**; bar = 20 μm), a centrohelid that harbors algal endosymbionts (**d**; bar = 20 μm), and the kleptoplastic dinoflagellate *N. aeruginosum* (**e**; Gymnodiniaceae; bar = 10 μm). For *N. aeruginosum*, images are shown of the source of the kleptoplast (i.e., the cryptomonad *Chroomonas* sp), cells during algal cell ingestion, with an enlarged kleptoplast, and during the digestion of the kleptoplast in this order. Figure panels are courtesy of Dr. Ryo Onuma (*N. aeruginosum*) and Mr. Kaoru Okada (the ameba and centrohelid).

including heat, cold, drought, and high-intensity light, increase photosynthetic oxidative stress[6–9]. Mitochondrial respiration also generates ROS[10]. As a result, eukaryotes had already developed several mechanisms to reduce ROS generation and quench ROS before acquiring the chloroplast, which were probably prerequisites and also contributed to eukaryogenesis[11]. However, some environmental stresses predominantly increase ROS generation through photosynthesis (e.g., high light, low $CO_2$, etc., as explained later) rather than through respiration[6–9]. Thus, algae and plants have evolved additional mechanisms to cope with ROS generation by

photosynthesis in addition to that by respiration[6–9,12]. These mechanisms have been extensively studied in algae and plants and are probably required for eukaryotes to establish chloroplasts and the ability to photosynthesize. However, this situation is not specific to organisms that possess innate chloroplasts but is also applicable to those engaged in acquired phototrophy. In addition, recent studies have shown that even unicellular predators are exposed to oxidative stress when they feed on algae under light conditions[13].

In this review, I briefly introduce recent advances in our understanding of the endosymbiotic evolution of photosynthetic

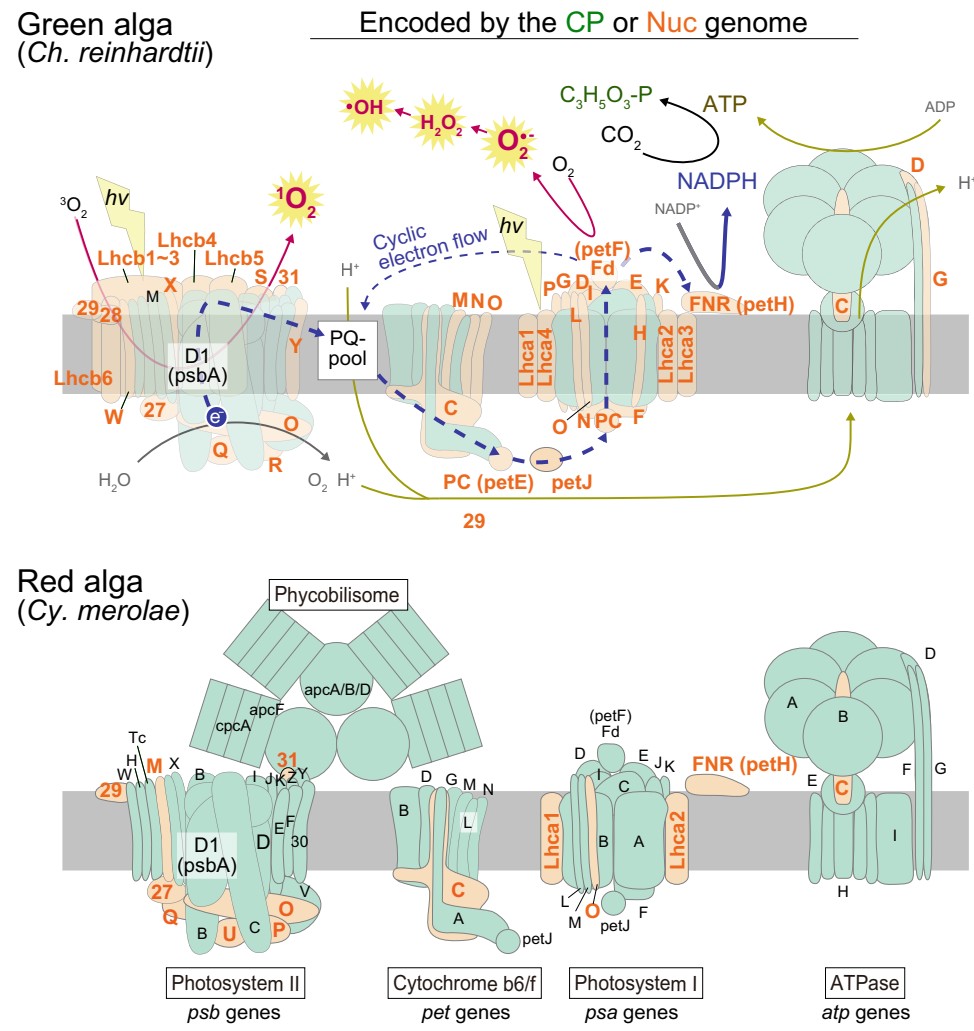

**Fig. 2 Composition of and ROS generation by the photosynthetic apparatus.** The illustration is modified from ref. [53]. and shows the photosynthetic apparatus in the green alga *Chlamydomonas reinhardtii* and the red alga *Cyanidioschyzon merolae*. Components encoded by the chloroplast and nuclear genomes are depicted in green and orange, respectively. In the green algal photosynthetic apparatus, the light-harvesting, electron flow, and ROS generation are also shown. In addition to the linear electron flow (thick blue dotted line) that generates ATP and NADPH and releases oxygen as a byproduct, the cyclic electron flow (thin blue dotted line) which generates ATP but does not produce NADPH or release oxygen is also depicted.

eukaryotes and the mechanisms employed by algae and plants to cope with photosynthetic oxidative stress. For further details on these topics, please refer to previous reviews cited in the respective sections. Then, I summarize and compare the knowledge regarding kleptoplasty, photosymbiosis, and unicellular predators feeding on algae with those in algae and plants. Through this comparison, it becomes evident that despite evolving independently in different lineages, there are several common features in mechanisms that have evolved to cope with photosynthesis-induced oxidative stress in eukaryotes that prey on algae, engage in acquired photosynthesis, or possess innate chloroplasts. Thus, there are certain evolutionary constraints for eukaryotes to develop mechanisms for coping with photosynthetic oxidative stress. Additionally, these mechanisms are likely gradually acquired by several eukaryotic lineages by incorporating photosynthetic cells (as prey, kleptoplasts, or endosymbionts) before establishing innate chloroplasts.

## Establishment and spread of chloroplasts in eukaryotes through endosymbiosis

Mitochondria originated from an alpha-proteobacterial endosymbiont that resided in a host cell derived from a group of archaea known as Asgard archaea. Several pieces of evidence indicate that the last eukaryotic common ancestor (LECA) already possessed mitochondria and many other eukaryotic traits, including complex membrane-bound compartments (nuclear envelope, ER, peroxisome, lysosome, Golgi apparatus, etc.) and complex cytoskeletal systems[11,14–16]. However, due to the lack of evolutionary intermediates between archaea and eukaryotes, the precise details of how LECA and mitochondria evolved are still a subject of debate. There are several hypotheses, essentially grouped into either the mito-early scenario, which considers the driving force of eukaryogenesis to be proteobacterial endosymbiosis into an archaeal host, or the mito-late scenario, in which an amitochondriate eukaryotic host ingested a proteobacterial endosymbiont[11,14–16].

After the LECA diverged into several eukaryotic lineages, the chloroplast was established in a unicellular ancestor of Archaeplastida through conversion of a cyanobacterial endosymbiont into a photosynthetic organelle in a process known as primary endosymbiosis; this later diverged into glaucophytes, red algae, green algae, and plants (Fig. 1a; note that the phylogenetic relationships of eukaryotes are still a matter of debate, especially for older branches)[3–5]. The origin of primary chloroplasts had been

estimated between 0.9 and 1.6 billion years ago[17]. However, a recent report of multicellular red algal fossils dating to around 1.6 billion years ago, along with molecular clock analysis, suggests an origin dating back to 1.9–2.1 billion years ago[18]. After the split of red and green algae, the primary chloroplast was subsequently spread into many lineages of eukaryotes as a complex chloroplast through secondary endosymbiotic events of red and green algae or even higher-order endosymbiosis, such as the endosymbiotic integration of an alga possessing a secondary chloroplast into another lineage of eukaryote (Fig. 1a)[3–5]. In addition, some lineages of dinoflagellate further replaced their original chloroplast of red algal origin with those of green algae, cryptophytes, haptophytes, or diatoms (Fig. 1b)[19]. After the secondary endosymbiotic event, the red or green algal contents other than the chloroplast were lost, except in the cryptophytes and chlorarachniophytes, both of which still retain a reduced nucleus (known as a nucleomorph) of red and green algal origin, respectively.

There is a wide consensus that two independent secondary symbiotic events involving distinct lineages of green algae gave rise to complex chloroplasts in euglenoids and chlorarachniophytes[4,5]. However, how many independent endosymbiotic events led to complex chloroplasts derived from red algae is still a matter of debate[4,5]. This is largely because phylogenetic analyses of chloroplast genes place complex chloroplasts of red algal origin in a monophyletic clade. However, analyses of nuclear genes divide the hosts to several different eukaryotic supergroups, with intervening heterotrophic lineages in which there is no evidence of there ever having been the chloroplast. Thus, it is unclear whether there were multiple independent endosymbioses of closely related red algae into various lineages of eukaryotes or if a secondary chloroplast of red algal origin was established once and then horizontally transferred to other lineages[4,5]. A recent hypothesis propose that the initial secondary endosymbiosis occurred in a heterotrophic ancestor of cryptophytes[20,21]. The cryptophyte secondary chloroplast is hypothesized to have been horizontally transferred to stramenopiles via a tertiary endosymbiotic event of a cryptophyte, and then to haptophytes through a quaternary endosymbiotic event of a stramenopile[20,21].

The question then arises as to how the cyanobacterial ancestor of the chloroplast entered the eukaryotic host cell, but this remains unresolved. Recent studies raised a possibility that the ancestor of Archaeplastida exhibited mixotrophic behavior by performing photosynthesis within chloroplasts while also feeding on other microorganisms by phagocytosis[3,4]. This is because, even today, a few lineages of the Archaeplastida, such as prasinophytes of green algae[22,23] and predatory flagellates named *Rhodelphis* (having lost photosynthetic activity but possessing relic plastids) that are closely related to red algae[24], are phagotrophic and feed on microorganisms for nutrition. In contrast to these minor examples, many other algae in Archaeplastida are not phagotrophic, suggesting that they lost their phagotrophic ability, if the hypothesis is correct. Whether an organism remains as a mixotroph or loses either its photosynthetic or phagotrophic abilities is suggested to depend on the balance between the cost of maintaining structures for photosynthesis and phagocytosis and the availability of light, the dissolved inorganic nutrients for photosynthetic growth, and prey in the given environment[25].

Although the ancestor of Archaeplastida might have been phagotrophic, it is still unclear whether the cyanobacterial ancestor of primary chloroplasts was also engulfed by the host through phagocytosis as often speculated[26]. The primary chloroplast is surrounded by two membranes: the inner and the outer envelope which probably correspond to the cytoplasmic and the outer membrane of cyanobacteria, respectively, although the latter was largely modified by the eukaryotic host[26,27]. Thus, if a cyanobacterium was engulfed through phagocytosis, it would be necessary to consider that it was initially enclosed by three membranes, and that the phagosomal membrane was lost later[26]. However, there are known examples in which infectious bacteria invade other cells, and in less common instances, bacterial endosymbionts reside either in other bacteria[15,28] or in the cytoplasm of nonphagocytic eukaryotic cells[29]. Thus, phagocytosis is not necessarily required for the establishment of intracellular symbiosis, although many lineages of extant eukaryotes ingest endosymbionts by phagocytosis[15,29]. This is also the case for the relationship between proteobacteria and mitochondria, both of which are surrounded by two membranes. While it is still controversial, a recent study suggests that phagocytosis emerged independently in various eukaryotic lineages after the appearance of LECA with mitochondria, which also suggests that the proteobacterium entered the host cell through a mechanism other than phagocytosis[28,30].

In contrast to the situation of the mitochondrion and primary chloroplast, several lineages possessing complex chloroplasts are phagomixotrophic, as observed in most euglenophytes, cryptophytes, chrysophytes in the stramenopiles, haptophytes, and dinoflagellates[25,31]. In addition, species engaging in acquired phototrophy (i.e., photosymbiosis and kleptoplasty) often exhibit a close evolutionary relationship with algae that possess innate complex chloroplasts (Fig. 1)[1,32,33]. Therefore, it is more likely that complex chloroplasts were established through the phagotrophic ingestion of eukaryotic algal prey, followed by their (either the whole algal cells or the chloroplasts/kleptoplasts) transient retention and eventual obligatory retention as permanent endosymbionts by unicellular eukaryotes[3–5,32–34]. Supporting this assumption, complex chloroplasts are surrounded by four (or three) membranes in which the inner two are derived from the inner and outer envelopes of the primary chloroplast. In the case of the complex chloroplast surrounded by four membranes, the two additional membranes are believed to correspond to the plasma membrane of the engulfed alga and the phagosomal membrane of the host cell, respectively, which are later connected or replaced by ER in some lineages[26,35] (see also the hypothesis explained in ref. [34]).

In summary, acquired phototrophy (as shown in Figs. 1, 3) may represent an intermediate stage between phagotrophic feeding on algae and the acquisition of innate chloroplasts through secondary or higher-order endosymbiosis. Nevertheless, it remains unclear how the bacterial ancestor of the primary chloroplast entered the eukaryotic host cell. Whereas acquisitions of photosynthetic organelle by primary endosymbioses were extremely rare events, with only the Archaeplastida and *Paulinella* (introduced later) cases known, acquisition of complex chloroplasts of eukaryotic algal origin occurred numerous times. The reason for this is still unclear. However, it might have been easier for eukaryotes to acquire and use chloroplasts that were already modified for eukaryotic use than to transform bacterial endosymbionts into chloroplasts through metabolic and genetic integration[5], which I introduce in the next section.

**Genetic and metabolic integration of chloroplasts into eukaryotic host cells.** During the integration process of a cyanobacterial endosymbiont (i.e., by primary endosymbiosis) or eukaryotic algal endosymbiont (by secondary or higher-order endosymbiosis) into a eukaryotic host cell as a chloroplast, numerous genes were lost from the endosymbiont, resulting in a loss of autonomy. This was also the case for the earlier conversion of the proteobacterial endosymbiont to the mitochondrion[15,36,37]. In the case of primary chloroplast establishment, the chloroplast genome was reduced to approximately 5% of the original

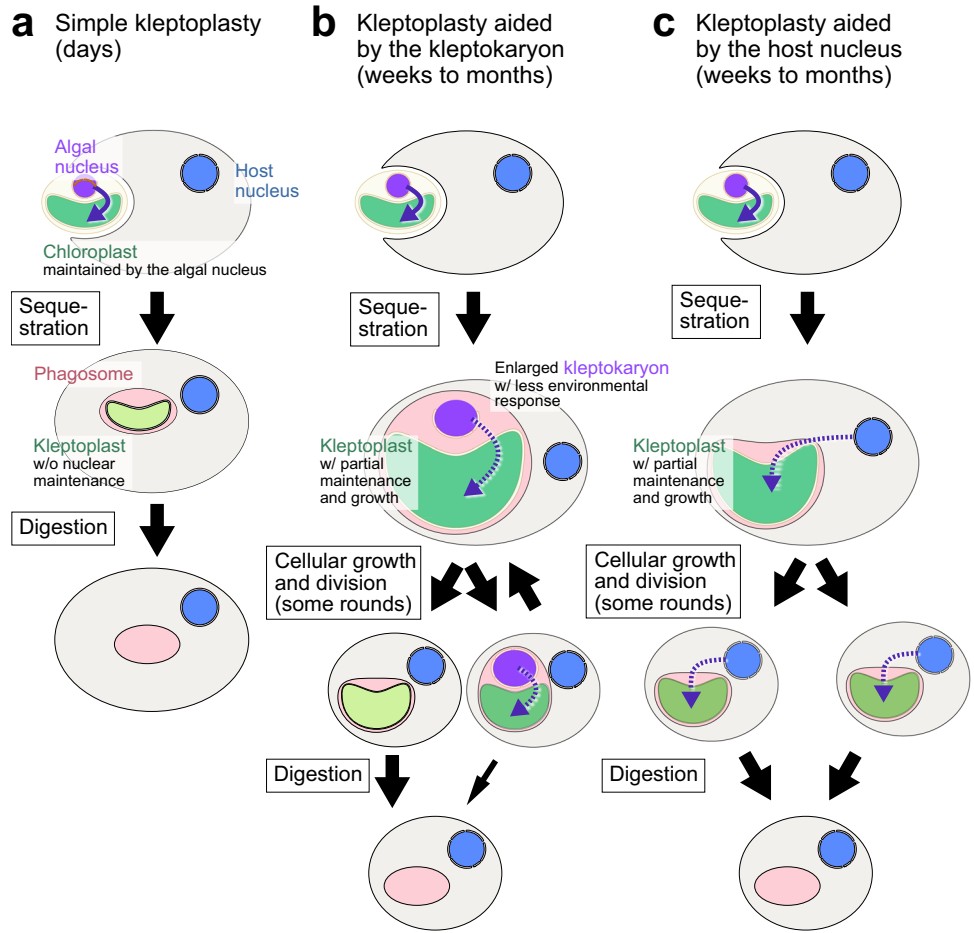

**Fig. 3 Several types of kleptoplasty. a** Simple kleptoplasty with faster turnover rates observed in several lineages. Here, only the chloroplast of the prey is sequestered and utilized. **b** Kleptoplasty with slower turnover rates, supported by a kleptokaryon, as observed in the ciliate *M. rubrum/major* and the dinoflagellate *N. aeruginosum*. Here, the kleptokaryon is inherited by only one of the two daughter cells. In cells that have lost the kleptokaryon, the kleptoplast ceases to grow and then undergoes digestion. **c** Kleptoplasty with slower turnover rates, supported by kleptoplast-targeted proteins that are encoded in the host nuclear genome, as observed in the dinoflagellates *Dinophysis* and RSD, and a recently found kleptoplastic euglenoid, *Rapaza viridis*. Here, the kleptoplast grows for a certain period in *Dinophysis*, as depicted in the figure, but does not grow in RSD.

cyanobacterial genome[3–5]. Simultaneously, certain genes crucial for functioning as a photosynthetic organelle were transferred to the nuclear genome of the eukaryotic hosts through endosymbiotic gene transfer (EGT). Proteins encoded by EGT genes are translated within the cytosol of the host cell and subsequently transported back to the chloroplast through a specialized protein translocon that spans the inner and outer chloroplast envelopes[3–5].

In the case of the primary chloroplast, nucleus-encoded, chloroplast-targeted proteins contain N-terminal transit peptides that are recognized by the translocon. For complex chloroplasts, these targeting proteins additionally contain N-terminal signal peptides that help guide them across the additional membranes surrounding the chloroplasts[38]. The translocon is an evolutionarily hybrid protein complex that combines cyanobacterial pore-forming proteins with components added by the eukaryotic host to give import specificity[3]. With respect to transit peptides, there is a hypothesis suggesting that they evolved from linear cationic α-helical peptides that exhibit antimicrobial activity by interacting with and modulating the permeability of the bacterial membrane[3,39]. While the composition and origin of the mitochondrial translocon are distinct from those of the chloroplast, the mitochondrion-targeting transit peptide is also similar to the linear cationic α-helical peptides[39].

EGT played a role in the genetic integration of an algal endosymbiont into the host cell, as mentioned above. However, nucleus-encoded chloroplast-targeted proteins are not exclusively derived from the cyanobacterial (or red/green algal for complex chloroplasts) ancestors of the chloroplast. In fact, many proteins originate from eukaryotic hosts, with some derived from pre-existing proteins in eukaryotes and others acquired through horizontal gene transfer (HGT) from other organisms during the establishment of the chloroplast[3–5]. For example, it is estimated that 7–15% of primary chloroplast proteins are derived from bacteria other than cyanobacteria through HGT. Similarly, genomic analyses of various lineages possessing complex chloroplasts derived from red algae (Fig. 1a) have revealed that many nucleus-encoded chloroplast proteins originate from green algae. In addition, genomic analyses of euglenoids and chlorarachniophytes, which possess complex chloroplasts derived from green algae (Fig. 1a), have estimated that approximately 30% and 50% of chloroplast proteins, respectively, are derived from red algae through HGT. These nucleus-encoded chloroplast proteins of HGT origin are hypothesized to have been acquired from previous cryptic endosymbionts (shopping-bag hypothesis) or may have been obtained from prey organisms (feeding activity is also believed to facilitate HGT from prey[40])[4], as has been suggested by genomic analyses of kleptoplastidic dinoflagellates

and predatory unicellular organisms, which will be introduced later.

For example, approximately half of the enzymes involved in the Calvin-Benson-Bassham (Calvin) cycle, which is responsible for carbon fixation in chloroplasts, are derived from the host rather than cyanobacteria[41]. Thus, even in metabolic pathways derived from cyanobacterial endosymbionts, the genes encoding the enzymes involved may not necessarily be of EGT origin. In addition, host-derived proteins (i.e., both those that were pre-existing in the host or were acquired through HGT) have significantly contributed to the establishment of mechanisms necessary for chloroplast establishment[3,5]. Those include some components of the protein translocon, as described above, as well as transporters responsible for translocating metabolites across the chloroplast envelope, which are predominantly of host origin[42,43]. Of these, sugar-phosphate/phosphate transporters play an especially crucial role in transferring photosynthesized sugar-phosphate to the host cell. Moreover, efficient removal of photosynthates from the chloroplast enables the Calvin cycle to continue fixing carbon, thereby preventing the production of ROS and photoinhibition[3], which is introduced further below.

**ROS generation and photoinhibition in Photosynthesis**. The photosynthetic apparatus uses light energy absorbed by photosynthetic pigments (such as chlorophylls and phycobilins) to split water molecules into protons, oxygen molecules, and electrons. These electrons are then transferred through the two photosystems, PSII and PSI, that generate reductant NADPH and create a proton gradient across the thylakoid membrane. This proton gradient is then used to generate ATP, and NADPH and ATP are in turn used to reduce carbon dioxide and produce sugar-phosphate (Fig. 2). However, this process also generates ROS, the level of which increases in response to environmental stress (Fig. 2). For example, under high light conditions, excessive energy absorbed by pigments is transferred to the oxygen molecule, generating singlet oxygen ($^1O_2$). In addition, if the number of electrons extracted from water exceeds the use capacity of the Calvin cycle, it results in a surplus of electrons. These overloaded electrons then react with oxygen, generating the superoxide anion radical ($O_2^{\bullet-}$), which then undergoes dismutation to form hydrogen peroxide ($H_2O_2$), subsequently generating the hydroxyl radical ($\bullet OH$) (Fig. 2)[6–9]. Other environmental stresses, such as heat, cold, and drought, can also elevate ROS production by lowering the activity of the Calvin cycle[6–9]. In addition, during photosynthesis, one of the Calvin cycle enzyme ribulose-1,5-bisphosphate carboxylase/oxygenase (Rubisco), which fixes $CO_2$ through its carboxylase activity, also reacts with oxygen through its oxygenase activity, generating phosphoglycolate. To convert this substance into a reusable form, photorespiration takes place, during which $H_2O_2$ is generated as a by-product in the peroxisome[6–9].

Although excessive ROS can damage biomolecules such as proteins, lipids, and DNA, low levels of ROS act as signaling molecules that are important regulators of cellular processes both in prokaryotes and eukaryotes[7,12,44,45]. Thus, the levels of ROS within the cell are tightly controlled to maintain a delicate balance between their signaling function and their potential cytotoxic effects. Imbalances between ROS production and scavenging mechanisms can lead to oxidative stress, resulting in cellular dysfunction and ultimately cell death[7,44,45]. Since the electron transfer chain in mitochondria also generates ROS through the transfer of electrons to oxygen[10], and ROS are generated in other metabolic processes, such as peroxisomal beta-oxidation[46] (suggested to have reduced ROS generation in the mitochondrial electron transfer chain by taking over a portion of β-oxidation for

the mitochondrial ancestor[47]), LECA should have already evolved systems to maintain ROS homeostasis across several cellular compartments[11] as some specific mechanisms are introduced later. Thus, upon acquisition of primary or complex chloroplasts, the eukaryotic hosts could have coped to some extent with ROS originating from photosynthesis by expanding or repurposing existing systems[11].

In green parts of land plants, photosynthesis produces a considerably higher amount of ROS compared to the mitochondrial respiration[6,9]. However, this is partly because, in plant mitochondria, the alternative oxidase, which is absent in vertebrates[48], partially uncouples electron transfer from proton pumping and ATP generation, thereby reducing ROS generation[6]. In addition, land plants have evolved to produce more ATP in their chloroplasts than in their mitochondria in the light[49]. However, the levels of ROS generation in respective cellular compartments change depending on the environment[9]. Thus, it is difficult to compare the relative pressure by ROS generation during the courses of establishment of the mitochondrion and the chloroplast[11]. Nevertheless, as mentioned earlier, in response to environmental stressors, such as high light or low $CO_2$, which primarily elevate ROS generation through photosynthesis[6–8], photosynthetic organisms have developed additional mechanisms to cope with oxidative stress, in addition to those inherited from their non-photosynthetic ancestors. These mechanisms will be introduced later.

In addition to oxidative stress, photosynthesis also faces the risk of photoinhibition[8,50]. The reaction center of PSII is highly susceptible to photodamage caused by light, particularly UV and blue light. This photodamage can occur even at low light levels and intensifies with increasing light intensity. Normally, the damaged D1 (PsbA) protein, which forms a part of the PSII reaction center (Fig. 2), is continuously degraded, removed, and replaced with newly synthesized protein. However, under high light conditions, the rate of photodamage exceeds the rate of this repair process. In addition, high light or other environmental stresses accelerate ROS production, as described earlier, which compromises the repair process and leads to a decrease in photosynthetic activity known as photoinhibition. In addition, delaying the repair process further enhances ROS generation[50]. The D1 protein is encoded by the chloroplast genome (Fig. 2); however, several proteins involved in the repair process are encoded in the nuclear genome[8,51]. Thus, the eukaryotic host cell must engage in PSII repair and cope with photosynthetic oxidative stress, the levels of which fluctuate according to environmental changes.

**Mechanisms to cope with photosynthetic oxidative stress in algae and plants**. Eukaryotic algae and plants (i.e., organisms possessing innate chloroplasts) have developed a variety of mechanisms to minimize the generation of ROS and effectively quench ROS in response to environmental fluctuations and the state of the photosynthetic apparatus. These include the mechanisms listed below[7,8,12].

(1) Excess light energy absorbed by the photosynthetic apparatus is dissipated as heat[8,12]. This mechanism involves carotenoids associated with the photosynthetic apparatus and is known as non-photochemical quenching or qE. It exhibits a rapid response, occurring on a timescale of seconds to minutes independently of changes in gene expression. qE originated in cyanobacteria, but was modified following the establishment of the chloroplast.

(2) The photosynthetic apparatus undergoes photoacclimation (Fig. 4)[52]. On time scales of hours to days, the photosynthetic apparatus exerts changes in macromolecular composition and

## Prey-predator

## Kleptoplasty

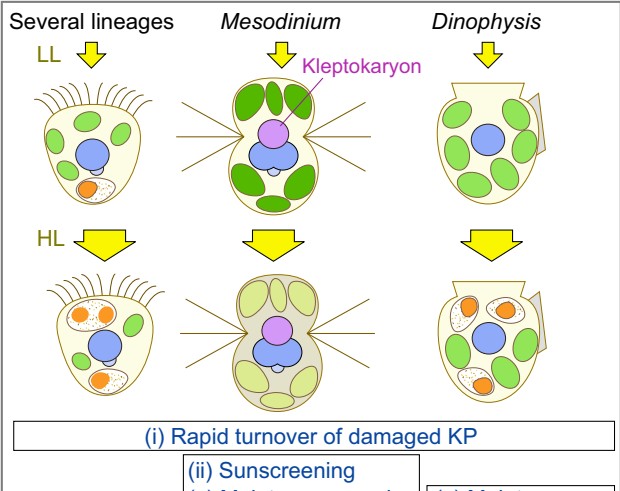

## Photosymbioses

## Algae and plants

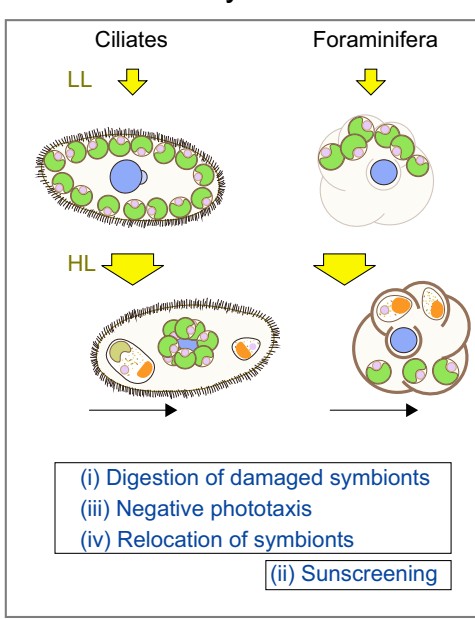

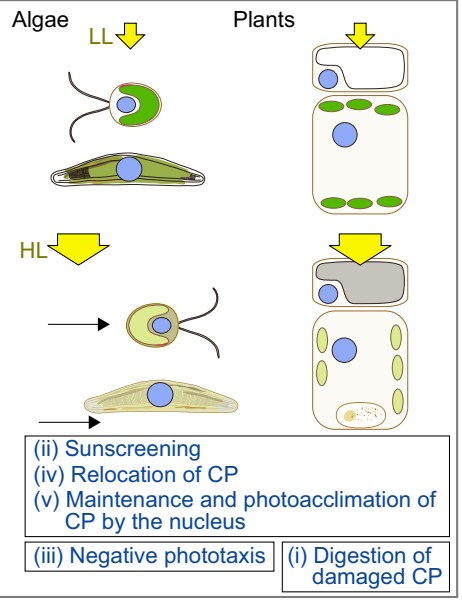

**Fig. 4 Comparison of mechanisms evolved in several types of eukaryotes to mitigate photosynthetic oxidative stress.** The figure highlights mechanisms other than ROS scavenging and compares them among eukaryotes that feed on algae (i.e., prey-predator interactions), engage in kleptoplasty or photosymbiosis, or possess innate primary or complex chloroplasts (i.e., algae and plants). Related mechanisms are assigned the same Roman numeral. Details and relevant references are provided in Supplementary Table 1. L, D, LL, HL, and KP indicate light, dark, low light, high light, and kleptoplasts, respectively.

ultrastructure in response to variations in light conditions. It does so by changing gene expression to maintain an optimal balance between light absorption and utilization. This includes changes in the abundance of light-harvesting pigment-protein complexes (i.e., light-harvesting antennas) and reaction center proteins. For example, under high light conditions, the size of light-harvesting antenna complexes is reduced to avoid the absorption of too much light energy. The mechanism of the photoacclimation originated in cyanobacteria. However, the composition of the antennas has been modified following the establishment of the chloroplast[53]. Cyanobacteria and red algae possess the phycobilisome as an antenna associated with PSII (Fig. 2). Unlike cyanobacteria, red algae possess the light-harvesting complex I

(LHCI) that is associated with PSI, similar to green algae and plants (Fig. 2). Green algae and plants lost phycobilisomes, and PSII acquired another light-harvesting complex (LHCII) (Fig. 2). LHC proteins are encoded in the nuclear genome (Fig. 2), and therefore changes in gene expression in the nuclear genome play a crucial role in photoacclimation in both organisms possessing primary and complex chloroplasts.

(3) Cells migrate to locations of appropriate light intensity through phototaxis[54] or relocate their chloroplasts[55] to adjust the level of light absorption (Fig. 4)[8,12]. Unicellular algae, whether possessing primary or complex chloroplasts, have the ability to sense light through photoreceptors such as cryptochromes and, in some cases, structures called eye spots, which provide directional

shading for the adjacent membrane-inserted photoreceptors[54]. In addition, the cells monitor their cellular redox state[56]. Then, they move toward areas with the appropriate light intensity[12,54]. Plants, being multicellular and immobile, sense light intensity using photoreceptors and then change the orientation of their leaves and use actomyosin to reposition chloroplasts, thereby adjusting the amount of light absorbed by the photosynthetic apparatus[55].

(4) Chloroplasts are shaded under high light conditions (Fig. 4)[8]. Under high light conditions, algae from several lineages, whether possessing primary or complex chloroplasts, produce sun-screening compounds such as mycosporine-like amino acids (whose subcellular localization remains unknown), which are also produced by cyanobacteria. In addition, plants exposed to high light accumulate anthocyanins and phenolic compounds in vacuoles of leaf epidermal cells to reduce the amount of UV and visible light reaching the photosynthetic apparatus.

(5) Many organisms, including non-photosynthetic prokaryotes and eukaryotes possess ROS scavenging enzymes such as superoxide dismutase, catalase, glutathione peroxidase, and peroxiredoxin and possess antioxidant molecules such as glutathione and carotenoids[7,57]. Cyanobacteria also possess numerous types of ROS scavengers including those mentioned above, alpha-tocopherol (vitamin E) and phenolic compounds[58]. Apparently, photosynthetic eukaryotes inherited ROS scavengers derived from non-photosynthetic eukaryotic ancestors and the cyanobacterial ancestor of the chloroplast. However, photosynthetic eukaryotes developed ascorbate peroxidases (APXs) as additional ROS scavenging enzyme[59]. In vascular plants, ascorbate is the most abundant water-soluble antioxidant, found at very high concentrations (5–20 mM), and there are several isoforms of APX that localize to the chloroplast, peroxisome, and cytosol. However, this situation is apparently specific to vascular plants because eukaryotic algae either possessing primary or complex chloroplasts contain much lower concentrations of ascorbate and possess only a single isoform of either chloroplastic or cytosolic APX[59].

(6) The systems for maintaining ROS homeostasis also exist in prokaryotes and non-photosynthetic eukaryotes, where sensory proteins respond to ROS concentrations above a certain threshold and activate the expression of ROS scavengers[44]. Thus, the ROS sensing system in photosynthetic eukaryotes probably evolved by adding new pathways for detecting the status of the chloroplast and light intensity to an existing network that non-photosynthetic ancestors had already developed.

In particular, photosynthetic eukaryotes either possessing primary or complex chloroplasts have evolved signaling systems that relay the light intensity sensed by photoreceptors, the redox state of the photosynthetic apparatus, and chloroplastic ROS levels to the nuclear genome[12]. These signaling systems allow cells to modulate the expression of nucleus-encoded ROS scavengers, components of the LHC, and transient photo-protective components of the photosynthetic apparatus such as early light-induced proteins (ELIPs) to mitigate photosynthesis-induced oxidative stress. Although ROS can directly or indirectly oxidize specific cysteine residue or Fe-S cluster of certain proteins and modulate their conformation and activity[44], the exact factors and mechanisms underlying the process of sensing photosynthesis status remain unknown.

A recent study showed that redox-sensitive cysteine residues, where R-SH is oxidized to R-SOH and further form intra- or inter-molecular disulfide bonds, were acquired by many nucleus-encoded proteins of EGT origin during the establishment of the primary chloroplast[60]. Furthermore, during the establishment of the complex chloroplast in diatoms, reactive cysteines were introduced into pre-existing proteins[60]. Another study showed

that diatoms evolved a unique set of a nucleus-encoded chloroplast-localized kinase and membrane protein to sense the status of the photosynthetic apparatus for photoacclimation[61]. Thus, stramenopiles, including diatoms, have evolved a mechanism that senses photosynthetic status, which differs from Archaeplastida that possess primary chloroplasts, during the acquisition of their complex chloroplasts of red algal origin (Fig. 1a). These findings emphasize that mechanisms for sensing the state of the chloroplast, such as redox regulatory networks, were important for photosynthetic eukaryotes to successfully cope with photosynthetic oxidative stress.

(7) The nuclear and chloroplast genomes cooperatively maintain and modulate the photosynthetic apparatus[32,53]. The chloroplast genome, although significantly reduced from the original cyanobacterial genome, encodes proteins necessary for genome maintenance, the expression of genome contents, and for intrinsic components of the photosynthetic apparatus, particularly those essential for photosynthesis, albeit with varying content depending on lineage (Fig. 2). On the other hand, the nuclear genome encodes peripheral subunits of the photosynthetic apparatus, some of which are dispensable in cyanobacteria during photosynthesis under low light and nutrient-rich conditions, such as PsbM, PsaE, PsaI, and PsaK, but they play a role in modulating photosynthetic efficiency under stress conditions, including PsbO, PsbP, PsbU, and PsaI (Fig. 2)[32,53]. The similar situation is also observed for the mitochondrion where proteins comprising the structural cores of the electron transport chain tend to be encoded within mitochondrial genomes across eukaryotes[36].

It has been suggested that genes, whose expression must be directly regulated by the redox state of their gene products or by electron carriers with which gene products interact, have been preserved in mitochondrial and chloroplast genomes to enable swift adjustment of their expression (co-location for redox regulation hypothesis). In fact, mRNA levels of chloroplast-encoded components of the photosynthetic apparatus change according to its redox level[37].

(8) Recent studies in the plant *Arabidopsis thaliana* have shown that drought stress, the acceleration of singlet oxygen generation in the chloroplast, or chloroplast damage caused by high light all result in the degradation of damaged chloroplasts in vacuoles through a specialized form of autophagy called chlorophagy (Fig. 4). These mechanisms protect cells and tissues by eliminating excessive ROS-producing chloroplasts and redistributing their nutrients to other cells[62]. Chlorophagy is reminiscent of mitophagy observed in animals, yeasts, and seed plants, where cells specifically remove dysfunctional or superfluous mitochondria to prevent uncontrolled ROS generation and energy losses[63,64].

As above, algae, whether possessing primary or complex chloroplasts, and plants have developed numerous mechanisms to cope with photosynthetic oxidative stress, and these probably have contributed to the success of photosynthetic organisms in fluctuating natural environments. However, it is noteworthy that similar mechanisms have also independently evolved in organisms that have acquired a photosynthetic organelle, distinct from chloroplasts, and engage in acquired phototrophy, as described below.

**Mechanisms to cope with photosynthetic oxidative stress in *Paulinella chromatophora*.** The thecate ameba *P. chromatophora* inhabiting freshwater environments (belonging to Rhizaria in Fig. 1a; the first axenic culture was reported in 2016[65]), along with two other sister species (the freshwater *P. micropora* and marine *P. longichromatophora*), is a unique example of a lineage that has

relatively recently (~0.1 billion years ago) integrated a cyano-bacterial endosymbiont as a photosynthetic organelle known as a chromatophore, independently from the establishment of the chloroplast (Fig. 1a). The cyanobacterium that became the chromatophore belongs to the alpha cyanobacteria which are different from the beta cyanobacteria from which the chloroplast evolved[3].

The chromatophore is surrounded by two membranes, which are separated by a thick peptidoglycan layer, similar to cyanobacteria[3]. The chromatophore genome has degenerated to about one-third of that of its ancestral cyanobacterium. One-third of the chromatophore proteome is encoded in the host nuclear genome, and these genes are translated in the cytoplasm before being transported to the chromatophore. As in chloroplasts, chromatophore-targeted proteins include those of both host and HGT origins and those of EGT origin.

Like the photosynthetic apparatus of the chloroplast, only the peripheral components (PsbN, PsaE, and PsaK) are encoded by the host nuclear genome[66]. In addition, genes encoding the family of high-light-inducible (Hli) photo-protective proteins, which transiently associate with the photosynthetic apparatus, have also relocated to the nuclear genome and increased their copy number[66]. Hli is closely related to ELIPs encoded in the nuclear genome in algae and plants. Hli has a single trans-membrane helix and is thought to be the ancestor of LHC proteins encoded in the nuclear genome of algae and plants, which have three trans-membrane helices. In *P. chromatophora*, *Hli* genes are upregulated either by ROS treatment or blue light. Thus, in *P. chromatophora* a signaling system similar to that found in algae and plants has evolved to adjust the photosynthetic machinery by transcriptional regulation of nuclear genes in response to the status of the chromatophore and the intensity of light[67].

In contrast to the Archaeplastida, the closest non-photosynthetic relatives for photosynthetic *Paulinella* spp. have been identified[3]. Of these, the non-photosynthetic *P. ovalis* feeds on bacteria, including cyanobacteria, and has been shown to possess at least two genes that were acquired through HGT from cyanobacteria[68]. These findings suggest that the establishment of the chromatophore in photosynthetic *Paulinella* resulted from the phagotrophic ingestion of a cyanobacterium and gene transfer from cyanobacterial and other microbial prey[3,68]. However, unlike phagotrophic *Paulinella*, photosynthetic species have lost the ability for phagocytosis, most likely due to living in nutrient-rich environments where phagotrophy is no longer essential[3].

As described above, the chromatophore has been partially genetically integrated into the host cell. However, photosynthetic *Paulinella* spp. are typically found in dim sedimentary habitats and are prone to bleaching and/or death under moderate to high light conditions, likely due to as yet incomplete integration between the host and the chromatophore[67].

**Mechanisms to cope with photosynthetic oxidative stress by unicellular organisms engaging in kleptoplasty.** Kleptoplasts are chloroplasts that are sequestered from phagocytosed algal prey and retained within a predator's cell for a certain period before being digested (Fig. 1e). The duration of kleptoplast retention varies, ranging from days to months, depending on both the host and the source of the kleptoplast (Fig. 3)[1,31]. When the klepto-plast remains intact, the photosynthates produced within it are transferred to the host cell. Eventually, during digestion, the contents of the kleptoplast are absorbed by the host cell. Klep-toplastic organisms must continuously replace kleptoplasts by ingesting fresh algal prey and thus are regarded as mixotrophs (Fig. 3). In some cases, along with the kleptoplasts, tran-scriptionally active nuclei known as kleptokaryons along with

other prey cell organelles are retained by the host cell (Fig. 3b). In addition to several lineages of unicellular eukaryotes, sacoglossan sea slugs also engage in kleptoplasty. However, in this review, only unicellular cases are introduced, and the case of sea slugs is summarized in other reviews[69]. Mechanisms used by klepto-plastic organisms to cope with photosynthetic oxidative stress include the following:

(1) Simple kleptoplasty in oligotrich ciliates and foraminifera (Fig. 3a). Oligotrichia are a group of ciliates that are widely distributed in both freshwater and marine environments and contain both heterotrophic and kleptoplastic species. Kleptoplas-tic species obtain kleptoplasts from various species of algal prey, requiring frequent reacquisition of prey chloroplasts[31,70]. These species do not appear to express genes related to kleptoplast maintenance and growth. In the case of *Strombidium* sp., which can stably proliferate by feeding on a chryptophyte prey in laboratory, when the algal prey is removed from the culture, more than 50% of cells survive under low light intensity for more than 4 days. This longevity under starvation is significantly longer than that observed in purely heterotrophic oligotrichs. However, under even moderate light intensity, more than 80% of cells die within two days after prey depletion, which more closely resembles the mortality rates of heterotrophic species[70]. In a similar manner, in some species of benthic foraminifera (described later) that acquire kleptoplasts from diatom prey, the digestion of kleptoplasts occurs earlier under light conditions than under dark conditions, and earlier under high light intensity compared to low light intensity[71,72]. This is also the case for other examples introduced below; however, the signals that trigger this acceleration of kleptoplast digestion are still unknown (though this point will be discussed later). The rapid declines in the longevity of the host cells and the high turnover of kleptoplasts under higher light intensity (Fig. 4) suggest a limited capacity for photodamage repair and photoacclimation in these kleptoplasts due to a lack of protein supply from the host cell, unlike organisms with innate chloroplasts.

(2) Kleptoplasty aided by kleptokaryons in the ciliate *Mesodinium rubrum/major* (Fig. 3b). *Mesodinium* is a genus of ciliates that are widespread in both marine and freshwater environments. In marine waters, six species have been identified: *M. pulex* and *M. pupula* are heterotrophic (phagotrophic), while *M. chamaeleon*, *M. coatsi*, *M. major*, and *M. rubrum* are kleptoplastic[73]. Among kleptoplastic species, the *M. rubrum/major* complex is widely distributed in coastal ecosystems and is known for causing nontoxic red tides[74]. *M. rubrum/major* specifically feeds on cryptophytes of the genera *Geminigera*, *Teleaulax*, and *Plagioselmis*. In contrast, other kleptoplastic species such as *M. coatsi* and *M. chamaeleon* exhibit greater flexibility and can utilize a wider range of cryptophyte species as sources of kleptoplasts[73].

*M. rubrum/major* (the first culture with prey being reported in 2004[75]) differs from the other kleptoplastic species in its unique mechanism of sequestrating the prey nucleus from the rest of the ingested algal prey. The kleptoplasts and other organelles, excluding the nuclei, are kept together as a distinct entity and a *M. rubrum* cell keeps around 20 kleptoplasts. One of the ingested prey nuclei is transported near the ciliate nucleus, where it enlarges and serves as a kleptokaryon (Fig. 4). The kleptokaryon remains transcriptionally active and expresses genes responsible for kleptoplast maintenance and growth[76,77].

While the kleptokaryon does not divide and is inherited by only one of the two daughter ciliates during cell division[78], its presence allows kleptoplasts to photoacclimate to some extent as those in the original algal cells and replicate for a few months before eventually being digested[77,79]. However, once the

kleptokaryon is lost, kleptoplasts start losing photosynthetic activity and are then digested[77]. In *M. rubrum*, the transcriptome of the ciliate nucleus does not contain any genes related to photosynthesis, suggesting that kleptoplast maintenance is primarily supported by the presence of the kleptokaryon[76,77]. However, unlike the nucleus of the original algal cell, the kleptokaryon is unable to change its transcriptome in response to changes in light conditions[76,77]. It is noteworthy that, when *M. rubrum* is exposed to high light, the ciliate cell itself produces MAAs to reduce the amount of light reaching the kleptoplasts (Fig. 4)[80]. This process is likely regulated based on information encoded in the ciliate nuclear genome.

In contrast to *M. rubrum/major*, in *M. chamaeleon*, kleptoplasts do not replicate and are highly susceptible to high light intensity[73]. Under such conditions, they quickly lose their photosynthetic activity and are digested. This susceptibility is probably due to their inability to repair photodamage and undergo proper photoacclimation.

(3) Kleptoplasty aided by kleptokaryons in the dinoflagellate *Nusuttodinium* (Fig. 3b)[81,82]. The genus *Nusuttodinium* (belonging to Gymnodiniaceae in Fig. 1e) contains both heterotrophic and kleptoplastic species. Among kleptoplastic species, *N. poecilochroum* inhabits seawater in beach areas and feeds on a wide range of cryptophytes as sources of kleptoplasts. While the kleptoplast is kept for about a week before being digested, the cryptomonad nucleus is digested within a few hours after ingestion. In contrast, the freshwater species *N. aeruginosum* (the first axenic culture with prey being reported in 2020[81]) specifically feeds on cryptomonads *Chroomonas* spp. and retains the algal nucleus (kleptokaryon) and other organelles in addition to kleptoplasts. Once ingested, the kleptokaryon undergoes endoreduplication, and the kleptoplast enlarges to a size more than 20-fold its original size; it can then survive for more than a month before being digested (Figs. 1e and 3b). In *N. aeruginosum*, the enlarged kleptokaryon is inherited by only one of the two daughter cells during cell division, similar to the case of the ciliate *M. rubrum* introduced above. In the cell that has lost the kleptokaryon, the kleptoplast ceases growth and is digested within three rounds of cell division[82]. Thus, kleptokaryons are responsible for maintaining photosynthetic activity and enabling the growth of kleptoplast. However, as in the case of *M. rubrum*, the enlarged kleptokaryon almost loses its transcriptional responses to light[81]. At this point, it remains unknown whether any genes in the nuclear genome of *N. aeruginosum* are also involved in the maintenance of kleptoplasts.

Although photoacclimation of the photosynthetic apparatus has not been examined, kleptoplasts and *N. aeruginosum* are both more susceptible to high light than free-living cryptomonad prey. However, the kleptokaryon apparently reduces the damage caused by high light, as it is evident from the fact that cells that have lost the kleptokaryon die more quickly than those that retained it[81].

(4) Partial genetic integration of kleptoplasts in the dinoflagellate *Dinophysis* (Fig. 3c). The dinoflagellate genus *Dinophysis* (belonging to Dinophysiales in Fig. 1b; the first culture with prey being reported in 2006[83]) is associated with diarrheic shellfish poisoning worldwide. *Dinophysis* spp. feed on the ciliates *M. rubrum/major* by using a peduncle to suck out the contents of the ciliate prey, whereby they obtain the ciliate's kleptoplasts, which originate from cryptomonads[83] as introduced above. *Dinophysis* spp. do not feed on cryptophytes directly and thus rely entirely on *Mesodinium* for a supply of kleptoplasts.

In *Dinophysis*, the kleptoplast remains photosynthetically active for up to 2 months after ingestion under low light conditions. However, its activity decreases within days under moderate and high light conditions, and the photodamaged photosynthetic apparatus is not repaired[84]. In addition, under higher light condition, the kleptoplasts start being digested earlier than under low light condition[85,86]. Unlike *M. rubrum/major*, the photosynthetic apparatus in *Dinophysis* does not photoacclimate to changes in light intensity[85]. However, *Dinophysis* is capable of undergoing several rounds of cell division with kleptoplasts continuing to grow and divide, even under prey-starved conditions[86], even though it does not retain the kleptokaryon[84].

Transcriptome analyses have revealed that the *Dinophysis* spp. nuclear genome encodes approximately 60 genes, the products of which are estimated to be imported and function within the kleptoplast[87,88]. These genes include enzymes that synthesize photosynthetic pigments, four APX proteins, and peripheral components of the photosynthetic apparatus, including PsbM, PsbO, PsbU, PetC, PetF, PetH, and a light-harvesting protein. Importantly, it should be noted that these genes do not originate from the cryptophyte that is the source of the kleptoplast. Instead, a majority of these are descended from a dinoflagellate ancestor, which possessed a complex chloroplast of red algal origin. Furthermore, some genes originate from the haptophytes, including genes related to those found in dinoflagellates with permanent chloroplasts of haptophyte origin (such as the Kareniaceae, shown in Fig. 1b). This suggests that the ancestors of extant *Dinophysis* engaged in haptophyte kleptoplasty at some point during their evolutionary history[88]. Nevertheless, *Dinophysis* possesses a limited number of chloroplast-related genes compared to other phototrophic algae possessing innate chloroplasts. This is probably one of the reasons why they are unable to sustain the chloroplast permanently.

(5) Partial genetic integration of kleptoplasts into the Ross Sea Dinoflagellate (RSD; Fig. 3c, but no growth of kleptoplast). RSD [belonging to the Kareniaceae (Fig. 1b); the first culture with prey being reported in 2007[89]] obtains the kleptoplast from the haptophyte *Phaeocystis antarctica* and is able to survive and maintain its kleptoplasts in the absence of prey for 5–30 months, but the kleptoplast does not grow within the dinoflagellate cell[89]. Although the growth of RSD relies on kleptoplasty, it has also retained a relic version of a non-photosynthetic complex plastid of red algal origin[90].

Transcriptomic analysis has revealed that the RSD nuclear genome has obtained many genes encoding kleptoplast-targeted proteins through HGT from various algal sources other than the haptophyte prey, many of which are shared with related species with fully integrated chloroplasts (e.g., *Karenia* and *Karlodinium*)[90]. This result suggests that, at least in some cases, a certain level of genetic integration can precede permanent organelle integration[90], which is also suggested by the recent study of a kleptoplastic euglenoid[91] However, this neither necessarily suggests that the genes encoding kleptoplast-targeted proteins were newly acquired for kleptoplasty, nor does it rule out the possibility that such kleptoplast-targeted proteins once functioned in the now-vanished chloroplasts[90,91]. The RSD nuclear genome encodes three components of the photosynthetic apparatus, PrtC, PetJ, and PsaE, which are also encoded in the nuclear genome of the prey *P. antarctica*[90]. However, no component of PSI is encoded in the RSD nuclear genome. This result is consistent with the observation of diminished PSII activity (~1/5) in the kleptoplast compared to the *P. antarctica* chloroplast[92] and suggests that the kleptoplast is mainly engaged in cyclic electron flow around PSI (Fig. 2) rather than a canonical linear electron flow involving PSII and PSI[90]. Based on these results and the fact that MAA—which screens UV and also functions as an antioxidant—are present in the kleptoplast (as

well as in the *P. antarctica* chloroplast)[92], the relatively long lifespan of the RSD kleptoplast may be attributable to both the reduced function of PSII, a major source of ROS, and the existence of MAA[90].

Collectively, these observations show that kleptoplasts tend to have longer lifespans when they are less prone to photodamage (e.g., under low light) or when they can be repaired with the assistance of a kleptokaryon or kleptoplast-targeted proteins encoded in the host nuclear genome. Moreover, kleptoplasts can only grow with the aid of a kleptokaryon or kleptoplast-targeted proteins encoded in the host nuclear genome (in the case of *Dinophysis* but not in RSD). In addition, when kleptoplasts experience enough cumulative damage, which is accelerated by exposure to high light conditions or the loss of a kleptokaryon, they are digested (Fig. 4). Thus, host cells can probably sense the status of kleptoplasts and digest damaged kleptoplasts to mitigate photosynthetic oxidative stress. This is similar to the situations of mitophagy and chlorophagy, as introduced above.

The relationship between a kleptoplast and a kleptokaryon is reminiscent of that between a complex chloroplast and a nucleomorph, as observed in cryptophytes, haptophytes, and certain dinoflagellates[93]. The nucleomorph can replicate and divide, and is inherited by daughter cells, unlike the kleptokaryon, which replicates but cannot divide and is inherited by only one of the daughter cells. However, if the replicated kleptokaryon could be segregated and inherited by both daughter cells, the period of kleptoplast retention would be extended. This is observed in the dinoflagellate *Durinskia baltica* (Peridiniales; Fig. 1b), which harbors a permanent diatom endosymbiont. During cell division, the nucleus of the diatom divides and is inherited by both daughter dinoflagellate cells[19].

**Mechanisms to cope with photosynthetic oxidative stress in unicellular organisms engaging in photosymbiosis.** Several lineages of unicellular and multicellular organisms, including animals such as Cnidaria, establish photosymbiotic relationships with unicellular algal endosymbionts (Fig. 1)[1,31]. Photosymbioses are mixotrophic, since the hosts also feed on other microorganisms[1,94]. In this relationship, the hosts provide waste products resulting from the consumption of microbial prey, such as nitrogen sources to the endosymbionts. In return, the endosymbionts supply the hosts with photosynthates. In many cases, both the host and symbiont are capable of independent proliferation in monocultures under suitable conditions. Aposymbiotic host cells can ingest endosymbiotic algal cells through phagocytosis. Then, the endosymbionts escape digestion and proliferate within the host cell while enclosed within a host membrane derived from the phagosomal membrane[95]. Therefore, endosymbionts maintain their integrity and autonomy, including the ability to cope with photosynthetic oxidative stress and photoinhibition within the host cell, even though the environmental conditions they encounter are determined by the host. Consistent with these situations, no examples of the genetic integration of endosymbionts into host cells, such as EGT or the import of host-nucleus-encoded proteins into an endosymbiont, has been found to date[96,97].

Although endosymbionts are accommodated by host cells in a non-invasive manner, algae are known to secrete organic compounds depending on the environment[98], which are likely utilized by host cells. For example, *Chlorella* (green alga) endosymbionts have been observed to secrete maltose under weakly acidic conditions, and this is then utilized by the host ciliate cell[95]. Here, two examples of well-studied photosymbioses observed in unicellular eukaryotes (i.e., ciliates and foraminifera)

are introduced. Recent studies on multicellular cases, such as corals[99] and hydras[100], can be found in other papers.

(1) Several species of ciliates accommodate the green alga *Chlorella* as endosymbionts. For example, the freshwater ciliate *Paramecium bursaria* harbors 300–500 *Chlorella* cells per individual in natural conditions, and each algal cell is enclosed by the perialgal vacuolar membrane[95].

In *P. bursaria*, the number of *Chlorella* endosymbionts per cell peaks under moderate light levels, and decreases under high light conditions (Fig. 4)[101]. When protein synthesis in the endosymbionts is inhibited in the light, thereby inhibiting the repair of photodamaged photosynthetic apparatus, *P. bursaria* digests them. This digestion does not occur in the dark or in light conditions in the presence of 3-(3, 4-dichlorophenyl)−1, 1-dimethylurea (DCMU), an inhibitor of photosynthetic electron flow[102]. In addition, *P. bursaria* reduces the number of endosymbionts per cell when the production of ROS by the endosymbionts is increased by high light exposure or by the application of an $O_2^-$ generator such as methylviologen[103]. Taken together, these results suggest that electron flow in the photosynthetic apparatus in which photodamage is not repaired generates high levels of ROS, and the host ciliate cell digests the endosymbionts to reduce oxidative stress (Fig. 4). However, it should be noted that blocking electron transfer from PSII to plastoquinone by DCMU is known to increase singlet oxygen generation[104], and thus, the results should be interpreted carefully while considering results of other experiments.

In addition, when exposed to high light, *P. bursaria* aggregates endosymbionts to shade both the host and the endosymbionts. However, under low light conditions it evenly distributes the endosymbionts beneath the cell membrane to maximize light uptake (Fig. 4)[105]. *P. bursaria* and the ciliate *Euplotes daidaleos* exhibit step-up (i.e., out-of-the-light) and step-down (i.e., into-the-light) photophobic responses to avoid abrupt changes in the intensity of light received (Fig. 4)[106]. The step-down reaction is dependent on the presence of *Chlorella* endosymbionts. In contrast, the step-up reaction is an intrinsic response of the ciliates themselves but is enhanced by the presence of endosymbionts[106].

(2) Foraminifera and Radiolaria are unicellular organisms that belong to the Rhizaria, and many species of these groups possess algal endosymbionts in seawater environments[1,31]. Foraminifera build calcium carbonate shells, using photosynthate produced by algal endosymbionts. The single cell is divided into multiple interconnected chambers (Fig. 4).

In accordance with changes in light quantity and quality, the benthic foraminifera *Operculina ammonoides* changes the number of diatom endosymbionts it contains[107]. Similarly, when exposed to high light or UV radiation, the benthic foraminifera *Amphistegina gibbose* digests diatom endosymbionts, resulting in the bleaching of foraminifera (Fig. 4)[108].

When exposed to high light, several species of benthic foraminifera calcify denser and thicker shells (Fig. 4)[109,110] and exhibit negative phototaxis, moving into shaded environments through pseudopodal locomotion (Fig. 4)[111]. In addition to this phototactic movement, when exposed to high light the benthic foraminifera *Marginopora vertebralis* relocates its dinoflagellate endosymbionts deeper into cavities within its shell using actin (Fig. 4)[112]. Both the phototactic movement and endosymbiont relocation do not occur in the presence of the photosynthetic inhibitor DCMU, suggesting that these responses depend on photosynthesis by the endosymbionts[111,112].

These findings suggest that, in photosymbioses, host cells have evolved several mechanisms to mitigate photoinhibition and oxidative stress. These mechanisms include the digestion of

severely photodamaged endosymbionts, phototaxis, the relocation of endosymbionts, and shading of endosymbionts under high light conditions (Fig. 4). In addition, host cells activate these responses by sensing light and the state of photosynthesis in the endosymbionts, similar to organisms that have established innate chloroplasts or engage in kleptoplasty.

**Mechanisms to cope with photosynthetic oxidative stress in unicellular organisms that feed on algae**. When unicellular transparent organisms feed on algae, light reaches the photosynthetic apparatus of the ingested prey during daylight hours, and ROS are then generated inside the predators (Fig. 1c). In addition, unregulated photosynthetic electron flow and the excitation of phototoxic free chlorophylls detached from the photosynthetic apparatus likely occur during digestion; both generate higher levels of ROS inside the predator. In fact, a recent study showed that numerous protists feeding on algae can detoxify chlorophyll to the catabolite cyclopheophorbide enol, which accumulates in cells and may act as an antioxidant[113]. However, the enzyme responsible for this reaction has not yet been identified. In another study of an ameba belonging to *Naegleria* in Discoba, when they fed on cyanobacteria, but not on non-photosynthetic prey, high light exposure led to oxidative stress and cell death. However, within 2 h, the cells acclimated to high light conditions through transcriptome changes, and surviving cells began to proliferate[13]. In addition, the amoebae reduced phagocytic uptake while accelerating the digestion of photosynthetic prey when shifted from darkness to light, which reduced the number of photosynthetic prey inside their cells (Fig. 4)[13]. This reaction presumably reduces ROS production until the cells can acclimate.

Upon illumination, amoebae (two Amoebozoa species and one Discoba species) feeding on cyanobacteria upregulate a homolog of genes involved in chlorophyll degradation/detoxification in plants and algae. This gene was acquired through HGT from photosynthetic organims, which are presumably their prey. In addition, genes involved in synthesis of carotenoids, which quench ROS, were also upregulated. These genes are also upregulated by light or the addition of ROS to the culture even when feeding on non-photosynthetic prey[13].

In other predators that feed on algae such as ciliates and heterotrophic dinoflagellates, there have been several reports of an increase in the ingestion and digestion rates of algal prey after acclimation to light conditions from dark conditions[114,115]. In the heterotrophic dinoflagellate *Noctiluca scintillans*, the acceleration of digestion was shown to occur in response to photosynthetic but not non-photosynthetic prey[114]. The increase in digestion rate observed in other predators is consistent with the results seen in *Naegleria* and is likely attributed to the rapid detoxification of algal prey. However, the advantages of the increase in the rate of prey ingestion after acclimation to light conditions remains unknown.

The faster digestion of photosynthetic prey in the light than in the dark is like the situation observed for organisms engaging in kleptoplasty or photosymbiosis (Fig. 1d, e), where they digest kleptoplasts or endosymbionts that accumulated photodamage; as a result, they probably generate higher levels of ROS in the light (Fig. 4). In addition, the above observations suggest that even several lineages of unicellular predators appear to have independently evolved mechanisms to sense light and state of prey through ROS and to change the way they handle photosynthetic prey to mitigate oxidative stress (Fig. 4).

**Conclusions and perspectives**. To date, research on photosynthetic microorganisms have been heavily biased toward organisms for which axenic cultures have been established, i.e., those that can grow only by photosynthesis[116]. On the other hand, for organisms showing acquired phototrophy and phagotrophy feeding on algae, recent advancements in culture systems and the integration of genomics have enable molecular-level analyses of both systems. Establishing these cultures is challenging due to the necessity of identifying suitable prey organisms and establishing a co-culture. Until a culture is successfully established, it is difficult to determine whether pigmented bodies within the cells are prey being digested, kleptoplasts, or endosymbionts. Thus, many organisms in this research field remain unanalyzed or even unrecognized. Indeed, a kleptoplastic organism was reported in euglenoids[91] and centrohelids[117] very recently.

While our understanding is still in its early stages, it is becoming evident that independent lineages engaged in acquired phototrophy have undergone parallel evolution of similar mechanisms to mitigate photoinhibition and cope with photosynthetic oxidative stress. These mechanisms include adjusting light intensity through shading, phototaxis, and the relocation of endosymbionts, as well as digesting damaged endosymbionts or kleptoplasts, likely in part through ROS signaling, as observed in organisms with innate chloroplasts (algae with primary and complex chloroplasts and plants) (Fig. 4; Supplementary Table 1). Here it should be noted that organisms possessing primary chloroplast or several different endosymbiotic origins of complex chloroplasts also developed similar mechanisms to cope with photosynthetic oxidative stress independently. Furthermore, at least a few mechanisms for coping with photosynthetic oxidative stress have evolved even in organisms feeding on algae and the associated genes may have been acquired by HGT from photosynthetic prey (Fig. 4; Supplementary Table 1). Currently, it is still unclear whether the cyanobacterial ancestor of the primary chloroplast was taken up through phagotrophy or by another mechanism, in contrast to the acquisition of complex plastids through secondary or higher order symbiosis, which most likely occurred through phagocytosis. In any case, some of the mechanisms for coping with photosynthetic oxidative stress observed in algae and plants today could have gradually developed even before the integration of native chloroplasts, when photosynthetic organisms existed within cells, whether as prey, kleptoplasts, or endosymbionts.

From an ecological perspective, it is now evident that acquired phototrophy is abundant and plays a significant role in aquatic food webs (Box 1). Furthermore, significant progress has been made in understanding the forms of associations (e.g., predators feeding on algae, photosymbiosis, kleptoplasty, and innate phototrophs) and their adaptability to different environments, considering the effects of environmental factors on photosynthetic oxidative stress and photoinhibition. Research has also begun to uncover the impact of these associations on other microorganisms and their environments (Box 1). These understandings suggest that the relationships with environments and other organisms also have influenced the process of endosymbiotic evolution.

Thus, to understand the processes of endosymbiotic evolution that have led to the establishment and spread of photosynthesis in eukaryotes, including the development of mechanisms to cope with photosynthetic oxidative stress, it is also important to understand the range of environments and fluctuations to which these organisms have been exposed. To this end, it may be effective for future studies to not only focus on cultivation methods that maximize growth using nutrient-rich media or abundant food sources but also to conduct cultivation and analysis under conditions that simulate their natural habitats.

---

**BOX 1 ▌ Relationship between ecosystems and the types of photosynthesis performed by unicellular eukaryotes**

Recent geobiographical studies showed that kleptoplastic ciliates account for ~40% of the total ciliate biomass in seawater[119]. In oligotrophic open oceans, photosymbiotic radiolaria dominate[119] and are estimated to have a mass equivalent to all other organisms[120]. In blooms, the kleptoplastic *M. rubrum/major* accounts for almost 100% of plankton biomass[94]. Thus, organisms of acquired phototrophy are substantial portion of all the biomass of aquatic ecosystems. Organisms conducting photosymbiosis especially dominate oligotrophic open ocean systems. In contrast, organisms conducting kleptoplasty, controlled by prey diversity and abundance, dominate in high-biomass areas[1]. However, kleptoplastic organisms with higher control and slower turnover rates, like *M. rubrum* and *D. acuminata*, have ecological distribution patterns closer to photosymbiotic species[119].

In oligotrophic environments, free-living algal biomass is limited by the scarcity of nutrients such as nitrogen, phosphorus, and iron. However, photosymbiosis facilitates efficient recycling of these nutrients, occasionally supplemented by bacterial prey, between the host and the algal endosymbiont, resulting in the production of higher amounts of organic materials than would typically be possible in such an environment[2]. In contrast, kleptoplasty generally involves nutrient flow in one direction from the algae to the host. The limited success of kleptoplasty with lower control and faster turnover rates in oligotrophic conditions may be attributed to the low availability of algal sources of kleptoplasts[2].

In aquatic ecology, protist functional groups are typically divided simply into phytoplankton (i.e., photoautotrophs) and microzooplankton (i.e., phago-heterotrophs). However, as mentioned, there is now an increasing recognition that many phytoplankton and microzooplankton are, in fact, mixotrophic, in that they are phagotrophic algae with innate chloroplasts or other organisms with acquired phototrophy[94,116]. Furthermore, the nature of mixotrophy can vary: for example, kleptoplasty with slower turnover rates and photosymbiosis, resemble phototrophs, while kleptoplasty with lower control and faster turnover rates, resemble phagotrophs. In addition, the same species of mixotrophic algae alter the balance between photosynthesis and phagotrophy depending on the environment[23].

Recent theoretical and culture studies have shown that, in addition to nutrients and prey availability, temperature and light intensity, determine the dominant type in an environment[33]. For example, in high-light, low-nutrient environments where photosynthesis decreases due to photoinhibition, algae feeding on bacteria dominate. In contrast, in low light, high-nutrient environments, pure heterotrophs and pure phototrophs prevail. High temperature stress also decreases photosynthesis due to photoinhibition, but mixotrophic organisms compensate by increasing the acquisition of organic nutrients through phagotrophy to sustain higher metabolic rates[33].

The behaviors of organisms with acquired phototrophy or feeding on algae also impact the environment and other microorganisms. For example, by feeding on algae, kleptoplasty can reduce algal abundance, thereby limiting competition with heterotrophic predators and reducing competition with photoautotrophic algae for inorganic nutrients that are required for photosynthetic growth[121]. A recent study suggests that light-enhanced prey digestion by unicellular predators reduces surface phytoplankton biomass but accumulates it at a certain depth in the water column, a phenomenon known as the deep chlorophyll maximum[115].

**Reporting summary**. Further information on research design is available in the Nature Portfolio Reporting Summary linked to this article.

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

## Acknowledgements

I am sincerely grateful to the anonymous reviewers for their insightful and constructive suggestions to improve the manuscript. I also thank Dr. Onuma (Kobe University, Japan) and Mr. Okada (National Institute of Genetics, Japan) for kindly sharing photos of microorganisms. Research in the laboratory of the National Institute of Genetics is supported by the Grants-in-Aid for Scientific Research from the Japan Society for the Promotion of Science (20H00477) and the Japan Science and Technology Agency (JST)-Mirai Program (22682397).

## Competing interests

The author declares no competing interests.
