## [Peer Review File · Communications Biology]

Reviewers' comments:

Reviewer #1 (Remarks to the Author):

Review for Communications Biology (Nature) – July 2023

Manuscript: COMMSBIO-23-2379

This review is also available as a word document (see the attached file).

Title: "Taming the perils of photosynthesis by eukaryotes: constraints on endosymbiotic evolution in aquatic ecosystems"

Author: Shin-ya Miyagishima.

Brief summary of the manuscript

In this review of how eukaryotic lineages acquire(d) photosynthesis by integrating cyanobacteria or eukaryotic algae, enabling autotrophy, the main focus is on how the cells cope with ensuing endogenous reactive oxygen species (ROS) that cause oxidative stress. Various mechanisms to mitigate oxidative stress, which embodies a set of evolutionary constraints, are presented. The author claims that several observations suggest that the existence of such constraints and the evolution of mechanisms to cope with photosynthetic oxidative stress may trace back to prey-predator relationships.

Overall impression of the work

Though this is a rather well-written manuscript (containing informative helpful figures), with interesting thoughts about the evolution of chloroplast integration and the pivotal role of ROS during these processes, I have some qualms about the overall vision presented here, where certain tacit assumptions are not sufficiently clear to the reader (as my use of the term "tacit" already indicates), and even more importantly, not questioned at all. These assumptions come from the fact that the author works mostly with red algae and kleptoplastic dinoflagellates. We all develop blind spots because of our own research subject and focus.

When thinking about eukaryotic development being shaped by internal ROS formation, the primary process is of course the PRIOR integration of the alpha-proteobacterium like organism which became the mitochondrion. The manuscript (not including references) does not mention mitochondria once! Discussing the evolution of chloroplast integration and the pivotal role of ROS during this development, without considering the restraints and solutions embodied in the eukaryotic cell that became involved in the first primary uptake of a cyanobacterium like organism, I consider a rather grave omission. Leaving out any mention, or discussion, of anti-oxidant measures resulting from the "mito-uptake" and their influence on later acquisitions seems difficult to defend. I also feel that the author misses a chance by not considering that this ROS adaptation upon endosymbiosis is a major, highly important, recurring theme in eukaryotic evolution. In light of all this there is still clear room for improvement, I think. However, I also want to stress that this is a scholarly achievement that integrates a lot of information: it was a pleasure to read and I learned a lot.

Specific comments, with recommendations for addressing each comment

General remarks before:

- Please try to integrate the prior uptake and hardwiring of the mitochondrion into LECA in the text (see above and below). You might e.g. first read the following publication, and consider whether you find the arguments presented there worthwhile:

Speijer, D., Hammond, M., and Lukeš, J. (2020) Comparing early eukaryotic integration of mitochondria and chloroplasts in the light of internal ROS challenges: timing is of the essence. *mBio*, 11, 3. doi: 10.1128/mBio.00955-20.
<https://journals.asm.org/doi/10.1128/mbio.00955-20>

- There is a tendency in the whole manuscript to automatically go from present-day mechanisms to earlier mechanisms. I think the author should sometimes more critically look at (and mention!) the possibility that earlier events followed different scenarios. For instance, was the first cyanobacterium really taken up by phagocytosis or not (btw, I am not saying it was not, but that alternatives might still be considered)? Compare e.g. line 45 and line 102-107 of the manuscript: "... was apparently indispensable for...". NB: There is a recent article which claims LECA was not phagotrophic. See: <https://academic.oup.com/gbe/article/14/6/evac079/6596370>.

- A lot of the interpretation rests on the use of the photosynthetic inhibitor DCMU, an inhibitor of photosynthetic electron flow. However, often such electron flow inhibitors can lead to more ROS formation, instead of less. Has this been checked and could you incorporate the reference in the article?

- The section "Relationship between ecosystems and the types of photosynthesis performed by unicellular eukaryotes" feels a bit extraneous to the rest of the article. Maybe this should be relegated to supplementary material?

Specific remarks; I will deal with these using line numbers. The author is of course free to disregard my recommendations (however, if I indicate that I consider a statement really erroneous, I would urge him to make the adjustment I propose).

Abstract, Line 23-25: "This suggests that chloroplasts were established through ingestion or transient/permanent retention of algal prey by unicellular eukaryotes." Alas, this is wrong! Algae have already taken up plastids/chloroplasts themselves. Change "algal prey" to "cyanobacteria". The same holds true for line 66: "...of algal predation..." should read "...of cyanobacterial predation...". See also line 103, etc, etc....

Line 174: "...levels of ROS act as signaling molecules that are important regulators of cellular processes 8,10,24." Maybe it should be pointed out here that this situation arose by adaptations necessary upon the merger of archaeon and the alpha-proteobacterial endosymbiont to be....

Line 178-180 gives the unnuanced impression that this is not quite debatable (comparison in plants, role of AOX in plants, long history of prior adaptations in eukaryotes, etc, etc). For instance, see (again): <https://journals.asm.org/doi/10.1128/mbio.00955-20>.

Line 222-223 (language issue): "...cases, structure called eye spot." Should read: "...cases, structures called eye spots."

Line 236-238: "However..."; is that statement really true? I am not convinced, but if the author is, please add references in support of the claim.

Line 277-279: Above, I stated "that this ROS adaptation upon endosymbiosis is a major, highly important, recurring theme in eukaryotic evolution." Here we have a nice example with autophagy itself possibly evolving as a form of mitophagy to get rid of oxidatively damaged mitochondria. In light of this the author might want to refer to such insights....

Line 283 (language issue): "...stress, and that these probably have contributed..." Should read: "...stress, and these probably have contributed..."

Line 289-290 (language issues): "...environments (belongs to Cercozoa in Rhizaria in Fig. 1A; the first axenic culture was reported in 2016 37), ..." Should read: "...environments (belonging to Cercozoa in Rhizaria in Fig. 1A; the first axenic culture was reported in 2016 37), ..." (NB: also, no full stop after 37). And reread the passage please: can you really use the description "unique" (the amoeba AND two sister species)? Check for wrongly used full stops everywhere (e.g. line 437).

Line 302-311: confusing: "Hli proteins" or just one Hli protein?
Line 330 (language issue): "...remains intact; the..." Should read: "...remains intact, the..."
Line 364 (language issue): Opening sentence "clumsy" (rewrite?) and "2004 47.) differs" : get rid of the full stop, please. Possible rewrite: "was reported" change to "being reported".
Line 387 (language issue): Same kind of mistake, "belonging" is correct here, not "belongs".
Line 392 (language issue): Idem, "being" instead of "was". I will not point out any of these instances, further. Check such constructs between parentheses everywhere (line 410, etc).
Line 394-395 (style): "...more than 20-fold greater than its original size". Change to: "...more than 20-fold its original size".
Line 436 (language issue): "integration in" or "integration into" not "to"....
Line 445-447: How sure is the author of this claim with all the coming and going of chloroplasts during these long evolutionary histories?
Line 560 (language issue): "...which is presumably..." Change to "...which are presumably..." ("algae" is plural).
Line 570: "However, the reason for the increase..." Don't you mean the "mechanism behind" instead of the "reason for"?
Line 648-649 (language): "Indeed, a kleptoplastic organism was recently reported in euglenoids very recently." Reread and rewrite please.

Figure 1: The evolutionary reconstruction/grouping of the eukaryotic groups used in figure 1a are "under fire" nowadays. I am not saying it should be replaced, but this fact might be mentioned in the legend.

BTW, I missed a list of acronyms used....

Reviewer #2 (Remarks to the Author):

This interesting review contains a considerable amount of natural history, particularly concerning the various ways that non-photosynthetic eukaryotes co-opt chloroplasts and sometimes other organelles from photosynthetic eukaryotes. Two central themes are suggested to unify this natural history. First, that all eukaryotes containing photosynthetic organelles, whether endogenous or borrowed, must cope with reactive oxygen species (ROS). Second, that the various taxa that co-opt chloroplasts from other eukaryotes provide insight into the original acquisition by eukaryotes of cyanobacterial symbionts. With regard to the former, while this can hardly be doubted, it is by no means a novel observation, and indeed this point has been made many times in the literature. With regard to the latter, I rather doubt that this is the case. Notice that these opportunistic eukaryotes are borrowing organelles from other eukaryotes. They are not taking up free-living cyanobacteria. Why would this be the case? Domesticating free-living cyanobacteria is likely highly challenging. With the exception of *Paulinella*, this has only occurred once in the entire history of life. Taking up a eukaryote with chloroplasts, in whole or part, is apparently much less challenging, and, as the author ably demonstrates, has occurred repeatedly. In other words, this is likely an "apples to oranges" comparison. Taking up free-living cyanobacteria and turning them into organelles likely involves entirely different evolutionary challenges than merely taking up already-functioning organelles. Thus, the manuscript provides very interesting natural history data, but readers may become lost in the details (e.g., in the kleptoplast section) and wonder how the details illuminate the central message. In the case of the origin of plastids, I am not sure that the details are at all relevant.

Other comments follow:

Line 45-47: "Therefore, it is hypothesized that chloroplasts were established through the phagotrophic ingestion of photosynthetic prey organisms, followed by their transient retention and eventual obligatory retention as permanent endosymbionts by unicellular eukaryotes": So, will existing

transient relationships become permanent?

Line 72: "enslavement" who enslaved whom? Was the host enslaved to take care of the symbiont? Might be best to avoid this term.

Line 167-168: "In addition, the number of electrons extracted from water exceeds the use capacity of the Calvin cycle, which results in a surplus of electrons." Should this be conditional: "If the number of electrons..., a surplus of electrons results."

Line 234-236: "Many organisms, including non-photosynthetic organisms, have evolved ROS scavenging enzymes such as superoxide dismutase, catalase, and peroxidases and possess antioxidant molecules such as peroxiredoxin, glutathione, and carotenoids." The wording is troublesome—it sounds like these enzymes evolved over and over again. Maybe delete "evolved"?

Line 343-348: "In the case of *Strombidium* sp., when algal prey is removed from the culture, more than 50% of cells survive under low light intensity for more than four days. This longevity under starvation is significantly longer than is observed in purely heterotrophic oligotrichs. However, under even moderate light intensity, more than 80% of cells die within two days after prey depletion, which more closely resembles the mortality rates of heterotrophic species." Can this species be successfully grown with the algal prey?

Line 348-351: "In a similar manner, in some species of benthic foraminifera (described later) that acquire kleptoplasts from diatom prey, the digestion of kleptoplasts occurs earlier under light conditions than under dark conditions, and earlier under high light intensity compared to low light intensity." Could ROS be a signal to digest the kleptoplasts?

Line 398-399: "Thus, kleptokaryons are responsible for maintaining photosynthetic activity and growing kleptoplast." Wording: "of the growing keptoplast"?

Line 409-410: "The dinoflagellate genus *Dinophysis* (belongs to *Dinophysiales* in Fig. 1b; the first culture with prey was reported in 2006 55.)...." Delete period.

Line 436-437: again, delete superfluous period

Line 478: "cnidaria" Cnidaria

I thank the reviewers for very important and constructive comments to improve the manuscript (COMMSBIO-23-2379). I have considered all of the comments carefully and made the following revisions, which are highlighted in the uploaded text file. The reviewers' comments were copied and underlined and I have described revisions directly beneath each comment. In the manuscript, the revised portions are indicated in red text.

Reviewer #1

Overall impression of the work

(1-1) Though this is a rather well-written manuscript (containing informative helpful figures), with interesting thoughts about the evolution of chloroplast integration and the pivotal role of ROS during these processes, I have some qualms about the overall vision presented here, where certain tacit assumptions are not sufficiently clear to the reader (as my use of the term “tacit” already indicates), and even more importantly, not questioned at all. These assumptions come from the fact that the author works mostly with red algae and kleptoplastic dinoflagellates. We all develop blind spots because of our own research subject and focus.

When thinking about eukaryotic development being shaped by internal ROS formation, the primary process is of course the PRIOR integration of the alpha-proteobacterium like organism which became the mitochondrion. The manuscript (not including references) does not mention mitochondria once! Discussing the evolution of chloroplast integration and the pivotal role of ROS during this development, without considering the restraints and solutions embodied in the eukaryotic cell that became involved in the first primary uptake of a cyanobacterium like organism, I consider a rather grave omission. Leaving out any mention, or discussion, of anti-oxidant measures resulting from the “mito-uptake” and their influence on later acquisitions seems difficult to defend. I also feel that the author misses a chance by not considering that this ROS adaptation upon endosymbiosis is a major, highly important, recurring theme in eukaryotic evolution. In light of all this there is still clear room for improvement, I think. However, I also want to stress that this is a scholarly achievement that integrates a lot of information: it was a pleasure to read and I learned a lot.

Thank you for spending valuable time to review the original manuscript. I also appreciate your constructive feedback and suggestions for improvement. In response to your feedback and that of another reviewer (some of which are fundamentally the same issues), I have made extensive revisions to the manuscript as outlined below. As a result, I believe that the manuscript has been significantly improved.

General remarks before:

(1-2) Please try to integrate the prior uptake and hardwiring of the mitochondrion into LECA in the text (see above and below). You might e.g. first read the following publication, and consider whether you find the arguments presented there worthwhile:

Speijer, D., Hammond, M., and Lukeš, J. (2020) Comparing early eukaryotic integration of mitochondria and chloroplasts in the light of internal ROS challenges: timing is of the essence. mBio, 11, 3. doi: 10.1128/mBio.00955-20. <https://journals.asm.org/doi/10.1128/mbio.00955-20>

Thank you for your very important and constructive comments. I have read the review and checked many other papers regarding how LECA and the mitochondrion were established and the relationship between the establishment of the mitochondrion and mechanisms for coping with the oxidative stress. I have added explanations about these topics to the manuscript, including the mechanisms that should have already existed in LECA for mitochondrial oxidative stress response. Furthermore, I have substantially revised the content of the manuscript to include explanations about whether the mechanisms for coping with photosynthetic oxidative stress are extensions of pre-existing mechanisms in eukaryotic cells (for mitochondrial ROS generation) or newly acquired ones.

The added or revised parts relevant to this major point are: L34, L39, L51, L76, L168, L183, L236, L239, L254, L313, L317, L326, L360, and L372.

(1-3) There is a tendency in the whole manuscript to automatically go from present-day mechanisms to earlier mechanisms. I think the author should sometimes more critically look at (and mention!) the possibility that earlier events followed different scenarios. For instance, was the first cyanobacterium really taken up by phagocytosis or not (btw, I am not saying it was not, but that alternatives might still be considered)? Compare e.g. line 45 and line 102-107 of the manuscript: "... was apparently indispensable for...". NB: There is a recent article which claims LECA was not phagotrophic. See: <https://academic.oup.com/gbe/article/14/6/evac079/6596370>.

Thank you very much for pointing out this very important matter. I agree that it is still unclear whether primary chloroplasts were acquired through phagocytosis. This issue is even more crucial and subject to extensive debate regarding the origin of mitochondria.

I have reviewed several sources and clarified this issue in the manuscript. It is more likely that secondary and higher ordered chloroplasts were acquired through phagocytosis, but regarding the primary chloroplast, it is still not unclear how it entered the eukaryotic cell. Therefore, I have revised

all the previously mentioned statements in the manuscript related to this matter.

The added or revised parts relevant to this major point are: Abstract, L113, L125, L153, L702.

(1-4) A lot of the interpretation rests on the use of the photosynthetic inhibitor DCMU, an inhibitor of photosynthetic electron flow. However, often such electron flow inhibitors can lead to more ROS formation, instead of less. Has this been checked and could you incorporate the reference in the article?

Thank you for pointing it out. I have added this point to the manuscript (L609) with a reference (Flors et al., 2006).

(1-5) The section “Relationship between ecosystems and the types of photosynthesis performed by unicellular eukaryotes” feels a bit extraneous to the rest of the article. Maybe this should be relegated to supplementary material?

Thank you for your suggestion. According to your comment, I have realized that this section is not so critical to the main point of this manuscript. However, I believe that the prevalence of acquired photosynthesis is not uncommon, and the fact that it constitutes a significant portion of biomass in the environment is important information. Additionally, I consider the relationship with the environment to be crucial for future research. Therefore, I have moved the section to Box 1.

(1-6) Abstract, Line 23-25: “This suggests that chloroplasts were established through ingestion or transient/permanent retention of algal prey by unicellular eukaryotes.” Alas, this is wrong! Algae have already taken up plastids/chloroplasts themselves. Change “algal prey” to “cyanobacteria”. The same holds true for line 66: “...of algal predation...” should read “...of cyanobacterial predation...”. See also line 103, etc, etc....

Thank you for pointing it out. I have totally rewritten the abstract to clarify that the cyanobacterial symbiont became a primary chloroplast, and subsequently, algal endosymbionts became secondary or higher-ordered chloroplasts. Additionally, as you pointed out in comment #1-4, the origin of primary chloroplasts from cyanobacterial symbionts is widely accepted, while it remains unclear whether they originated from cyanobacterial prey. Therefore, I have made corresponding revisions in the abstract (and other parts of the text, as explained in my response to comment #1-4) to address this issue.

(1-7) Line 174: "...levels of ROS act as signaling molecules that are important regulators of cellular processes 8,10,24." Maybe it should be pointed out here that this situation arose by adaptations necessary upon the merger of archaeon and the alpha-proteobacterial endosymbiont to be....

Thank you for your constructive suggestion, as in my response to comment #1-2, I have added the situation between LECA and the mitochondrion, which precedes the acquisition of primary and complex chloroplasts, to the text where appropriate. The added parts relevant to this particular suggestion (#1-7) are: L51, L236, L239, and L326.

(1-8) Line 178-180 gives the unnuanced impression that this is not quite debatable (comparison in plants, role of AOX in plants, long history of prior adaptations in eukaryotes, etc, etc). For instance, see (again): <https://journals.asm.org/doi/10.1128/mbio.00955-20>.

Thank you for your important suggestion. I have reviewed various sources, and as you suggested, and I have realized the important point you raised and added a passage to the main text to explain this point clearly (L247). I have also provided explanations in the text regarding the expectation that LECA (L239) had already developed ROS response capabilities and the matter concerning AOX (L247).

(1-9) Line 222-223 (language issue): "...cases, structure called eye spot." Should read:): "...cases, structures called eye spots."

I have revised the text as you suggested. I also added an explanation about eye spots.

(1-10) Line 236-238: "However..."; is that statement really true? I am not convinced, but if the author is, please add references in support of the claim.

Thank you for pointing this out. After researching several reviews, I have realized that, apart from land plants containing high levels of ascorbate and glutathione, it is challenging to make clear phylogenetic comparisons. Therefore, I have extensively rewritten the section (L313-325).

(1-10) Line 277-279: Above, I stated “that this ROS adaptation upon endosymbiosis is a major, highly important, recurring theme in eukaryotic evolution.” Here we have a nice example with autophagy itself possibly evolving as a form of mitophagy to get rid of oxidatively damaged mitochondria. In light of this the author might want to refer to such insights....

Thank you for the constructive suggestion. I have added the explanation of mitophagy to the text (L372).

(1-11) Line 283 (language issue): “...stress, and that these probably have contributed...” Should read: “...stress, and these probably have contributed...”

I have made the revisions as you suggested (L377).

(1-12) Line 289-290 (language issues): “...environments (belongs to Cercozoa in Rhizaria in Fig. 1A; the first axenic culture was reported in 2016 37.), ...” Should read: “...environments (belonging to Cercozoa in Rhizaria in Fig. 1A; the first axenic culture was reported in 2016 37), ...” (NB: also, no full stop after 37). And reread the passage please: can you really use the description “unique” (the amoeba AND two sister species)? Check for wrongly used full stops everywhere (e.g. line 437).

I have fixed these errors throughout the text.

(1-13) Line 302-311: confusing: “Hli proteins” or just one Hli protein?

Thank you for pointing this out. Since there are multiple Hli variants in a single species of organism, I have explicitly mentioned “the family of Hli proteins”.

(1-14) Line 330 (language issue): “...remains intact; the...” Should read: “...remains intact, the...”

I have fixed the error (L426).

(1-15) Line 364 (language issue): Opening sentence “clumsy” (rewrite?) and “2004 47.) differs” : get

rid of the full stop, please. Possible rewrite: “was reported” change to “being reported”. Line 387 (language issue): Same kind of mistake, “belonging” is correct here, not “belongs”. Line 392 (language issue): Idem, “being” instead of “was”. I will not point out any of these instances, further. Check such constructs between parentheses everywhere (line 410, etc).

I have fixed these problem (L462, L485, L489, and other similar parts).

(1-16) Line 394-395 (style): “...more than 20-fold greater than its original size”. Change to: “...more than 20-fold its original size”. Line 436 (language issue): “integration in” or “integration into” not “to”....

I have fixed these errors (L492 and 533).

(1-17) Line 445-447: How sure is the author of this claim with all the coming and going of chloroplasts during these long evolutionary histories?

Thank you for the comment. I have added an explanation of the interpretation of the results (L545).

(1-18) Line 560 (language issue): “...which is presumably...” Change to “...which are presumably...” (“algae” is plural).

I have fixed the error (L661).

(1-19) Line 570: “However, the reason for the increase...” Don’t you mean the “mechanism behind” instead of the “reason for”?

I have changed that to “advantages for” (L671).

Line 648-649 (language): “Indeed, a kleptoplastic organism was recently reported in euglenoids very recently.” Reread and rewrite please.

I have fixed the error and added a new finding in centrohelids (L690).

(1-20) Figure 1: The evolutionary reconstruction/grouping of the eukaryotic groups used in figure 1a are “under fire” nowadays. I am not saying it should be replaced, but this fact might be mentioned in the legend.

BTW, I missed a list of acronyms used....

Thank you for pointing this out. I have added the point to the text (L89, L99) and the figure legend (1004). I have also revised Fig. 1 A according to the most recent hypotheses.

I apologize for any inconvenience, but this journal does not permit the inclusion of an abbreviation table, so I did not prepare one.

Reviewer #2

(2-1) This interesting review contains a considerable amount of natural history, particularly concerning the various ways that non-photosynthetic eukaryotes co-opt chloroplasts and sometimes other organelles from photosynthetic eukaryotes. Two central themes are suggested to unify this natural history. First, that all eukaryotes containing photosynthetic organelles, whether endogenous or borrowed, must cope with reactive oxygen species (ROS). Second, that the various taxa that co-opt chloroplasts from other eukaryotes provide insight into the original acquisition by eukaryotes of cyanobacterial symbionts.

With regard to the former, while this can hardly be doubted, it is by no means a novel observation, and indeed this point has been made many times in the literature.

With regard to the latter, I rather doubt that this is the case. Notice that these opportunistic eukaryotes are borrowing organelles from other eukaryotes. They are not taking up free-living cyanobacteria. Why would this be the case? Domesticating free-living cyanobacteria is likely highly challenging. With the exception of Paulinella, this has only occurred once in the entire history of life. Taking up a eukaryote with chloroplasts, in whole or part, is apparently much less challenging, and, as the author ably demonstrates, has occurred repeatedly. In other words, this is likely an “apples to oranges” comparison. Taking up free-living cyanobacteria and turning them into organelles likely involves entirely different evolutionary challenges than merely taking up already-functioning organelles.

Thus, the manuscript provides very interesting natural history data, but readers may become lost in the details (e.g., in the kleptoplast section) and wonder how the details illuminate the central message. In the case of the origin of plastids, I am not sure that the details are at all relevant.

Thank you for spending valuable time to review the original manuscript. I also appreciate your constructive feedback and suggestions for improvement. In response to your feedback and that of another reviewer (some of which are fundamentally the same issues), I have made extensive revisions to the manuscript as outlined below. As a result, I believe that the manuscript has been significantly improved.

The purpose of this manuscript is not to discuss the history of chloroplast acquisition by eukaryotes but to summarize, while providing and comparing examples, that some of the mechanisms for coping with photosynthetic oxidative stress, similar to those developed by organisms with true chloroplasts (algae and plants), have independently evolved in groups of organisms engaging acquired photosynthesis. In order to make this point clearer, I have rewritten the abstract, made some modifications to the existing L65-L73 (at the end of the Introduction section), and added a sentence to the Conclusion section (L693-707) to clarify the purpose of this manuscript.

I agree with your very important point, so I have removed all the sentences that might suggest that secondary (or higher-order) endosymbiosis, predation, or acquired photosynthesis reflect the initial chloroplast establishment process through primary endosymbiosis. Additionally, I have added a note highlighting the fundamental differences between the process of converting cyanobacterial chloroplasts and the acquisition of pre-existing chloroplasts by a different lineage of eukaryotic cells (L155-162).

Other comments follow:

(2-2) Line 45-47: “Therefore, it is hypothesized that chloroplasts were established through the phagotrophic ingestion of photosynthetic prey organisms, followed by their transient retention and eventual obligatory retention as permanent endosymbionts by unicellular eukaryotes”: So, will existing transient relationships become permanent?

Thank you for pointing this out. As you and reviewer #1 have pointed out and as described in the response to your comment #2-1, since it is unclear whether this argument is correct, I have removed it from the main text and abstract.

(2-3) Line 72: “enslavement” who enslaved whom? Was the host enslaved to take care of the symbiont? Might be best to avoid this term.

Thank you for the comment. I have omitted “enslavement” and revised that part as “through conversion of a cyanobacterial endosymbiont into a photosynthetic organelle” (L 87).

(2-4) Line 167-168: “In addition, the number of electrons extracted from water exceeds the use capacity of the Calvin cycle, which results in a surplus of electrons.” Should this be conditional: “If the number of electrons...., a surplus of electrons results.”

Thank you for the suggestion, I have revised the sentence as suggested (L224).

(2-5) Line 234-236: “Many organisms, including non-photosynthetic organisms, have evolved ROS scavenging enzymes such as superoxide dismutase, catalase, and peroxidases and possess antioxidant molecules such as peroxiredoxin, glutathione, and carotenoids.” The wording is troublesome—it sounds like these enzymes evolved over and over again. Maybe delete “evolved”?

Thank you for your suggestion. I have changed “have evolved” to “possess” (L313).

(2-6) Line 343-348: “In the case of Strombidium sp., when algal prey is removed from the culture, more than 50% of cells survive under low light intensity for more than four days. This longevity under starvation is significantly longer than is observed in purely heterotrophic oligotrichs. However, under even moderate light intensity, more than 80% of cells die within two days after prey depletion, which more closely resembles the mortality rates of heterotrophic species.” Can this species be successfully grown with the algal prey?

Thank you for the comment. Yes, the stable culture of Strombidium sp. has been established. I have added the information in L440.

(2-7) Line 348-351: “In a similar manner, in some species of benthic foraminifera (described later)

that acquire kleptoplasts from diatom prey, the digestion of kleptoplasts occurs earlier under light conditions than under dark conditions, and earlier under high light intensity compared to low light intensity.” Could ROS be a signal to digest the kleptoplasts?

Thank you for pointing this out. Currently the signal is not identified. Regarding this point, I have added explanations (L447) and possibilities (L563-565; L695~696).

(2-8) Line 398-399: “Thus, kleptokaryons are responsible for maintaining photosynthetic activity and growing kleptoplast.” Wording: “of the growing keptoplast”?

Line 409-410: “The dinoflagellate genus Dinophysis (belongs to Dinophysiales in Fig. 1b; the first culture with prey was reported in 2006 55.)....” Delete period.

Line 436-437: again, delete superfluous period

Line 478: “cnidaria” Cnidaria

I have fixed these errors (L497, L506, L534, and L577).

I believe that the manuscript has been substantially improved, thanks to the reviewers. I look forward to your feedback.

Yours sincerely,

Shin-ya Miyagishima

Reviewers' comments:

Reviewer #1 (Remarks to the Author):

Review for Communications Biology (Nature) – October 2023

Manuscript: COMMSBIO-23-2379_Revision

Title: "Taming the perils of photosynthesis by eukaryotes: constraints on endosymbiotic evolution in aquatic ecosystems"

Author: Shin-ya Miyagishima.

Brief summary of the manuscript (unchanged).

In this review of how eukaryotic lineages acquire(d) photosynthesis by integrating cyanobacteria or eukaryotic algae, enabling autotrophy, the main focus is on how the cells cope with ensuing endogenous reactive oxygen species (ROS) that cause oxidative stress. Various mechanisms to mitigate oxidative stress, which embodies a set of evolutionary constraints, are presented. The author claims that several observations suggest that the existence of such constraints and the evolution of mechanisms to cope with photosynthetic oxidative stress may trace back to prey-predator relationships.

Overall impression of the revision.

The author has clearly spent a lot of time to (partially) rewrite the original manuscript. I appreciate his detailed feedback in the rebuttal and the clearly indicated altered sections. He has made extensive revisions to the manuscript and as a result, I find the manuscript to be very significantly improved (and it was rather good to begin with). I am left with a few (very minor) points in the rewritten sections, which the author can deal with or not (see below). I congratulate him with an impressive overview of (t)his exciting field of research and the successful integration in the broader context of early eukaryotic evolution.

Specific comments, some with recommendations for addressing them.

Specific remarks are dealt with using line numbers. As I said, the author is free to disregard my recommendations.

Abstract, Line 30-31: "Additionally, the mechanisms in algae and plants had likely been gradually acquired by incorporating photosynthetic cells before establishing innate chloroplasts." I am not completely happy with this manner of formulating things at the conclusion of the abstract, though I do not consider the statement erroneous. But, it does not refer to: (i) pre-existing mechanisms to cope with ROS coming from "taming the mitochondrion" and (ii), less importantly, possible differences between the primary and "higher order" chloroplast origins. These are of course extensively dealt with in the main text.

Line 76: "Mitochondria originated from alpha-proteobacterial endosymbiont that resided in a host cell..." Either: "Mitochondria originated from an alpha-proteobacterial endosymbiont that resided in a host cell..." or "Mitochondria originated from alpha-proteobacterial endosymbionts that resided in a host cell..."

Line 156/157: "Whereas acquisition of photosynthetic organelle by primary endosymbioses are extremely rare in history, with only..." Same kind of issue: "Whereas acquisition of photosynthetic

organelles by primary endosymbioses are extremely rare in history, with only..." And maybe change "rare in history" to "rare events".

Line 161: "...transform bacterial endosymbiont into chloroplasts..." Same kind of issue: "...transform bacterial endosymbionts into chloroplasts..."

Line 244-246: "Thus, upon acquisition of primary or complex chloroplasts, the eukaryotic hosts could have to some extent cope with ROS originating from photosynthesis by expanding or repurposing existing systems." Should read: "Thus, upon acquisition of primary or complex chloroplasts, the eukaryotic hosts could have coped to some extent with ROS originating from photosynthesis by expanding or repurposing existing systems."

Line 318/319: "from non-photosynthetic eukaryotic ancestor and cyanobacterial ancestor of the primary chloroplast." Should read: "from non-photosynthetic eukaryotic ancestors and cyanobacterial ancestors of the primary chloroplast."

Line 324: "...much lower concentration of ascorbate..." Should read: "...much lower concentrations of ascorbate..." I'll leave out further examples of this specific grammatical error.

Line 699-701: "Furthermore, at least a few mechanisms for coping with photosynthetic oxidative stress have evolved even in organisms feeding on algae and that the associated genes may have been acquired by HGT from photosynthetic prey (Fig. 4; Table S1)." Should read: "Furthermore, at least a few mechanisms for coping with photosynthetic oxidative stress have evolved even in organisms feeding on algae and the associated genes may have been acquired by HGT from photosynthetic prey (Fig. 4; Table S1)."

Reviewer #2 (Remarks to the Author):

The author has made a thorough and conscientious effort to meet all of the reviewers' suggestions. I think this is a fine, information-rich review. I mention two very minor additional suggestions:

Line 23: "Recent studies have shown that, besides algae and plant with innate chloroplasts,": plants?

Line 38: It might be worth noting that Sanchez-Baracaldo et al. (2017, PNAS 114, E7737) suggest a much younger age for the origin of chloroplasts.

I thank the reviewers for additional important comments to improve the revised version of the manuscript (COMMSBIO-23-2379A). I have considered all of the comments carefully and made the following revisions, which are highlighted in the uploaded text file. The reviewers' comments were copied and underlined and I have described revisions directly beneath each comment. In the manuscript, the revised portions are indicated in red text.

Reviewer #1

(1-1) Brief summary of the manuscript (unchanged).

In this review of how eukaryotic lineages acquire(d) photosynthesis by integrating cyanobacteria or eukaryotic algae, enabling autotrophy, the main focus is on how the cells cope with ensuing endogenous reactive oxygen species (ROS) that cause oxidative stress. Various mechanisms to mitigate oxidative stress, which embodies a set of evolutionary constraints, are presented. The author claims that several observations suggest that the existence of such constraints and the evolution of mechanisms to cope with photosynthetic oxidative stress may trace back to prey-predator relationships.

Overall impression of the revision.

The author has clearly spent a lot of time to (partially) rewrite the original manuscript. I appreciate his detailed feedback in the rebuttal and the clearly indicated altered sections. He has made extensive revisions to the manuscript and as a result, I find the manuscript to be very significantly improved (and it was rather good to begin with). I am left with a few (very minor) points in the rewritten sections, which the author can deal with or not (see below). I congratulate him with an impressive overview of (t)his exciting field of research and the successful integration in the broader context of early eukaryotic evolution.

Specific comments, some with recommendations for addressing them.

Specific remarks are dealt with using line numbers. As I said, the author is free to disregard my recommendations.

Thank you once again for taking your valuable time. I have addressed all the comments received as follows.

(1-2) Abstract, Line 30-31: "Additionally, the mechanisms in algae and plants had likely been gradually acquired by incorporating photosynthetic cells before establishing innate chloroplasts." I am not completely happy with this manner of formulating things at the conclusion of the abstract, though

I do not consider the statement erroneous. But, it does not refer to: (i) pre-existing mechanisms to cope with ROS coming from “taming the mitochondrion” and (ii), less importantly, possible differences between the primary and “higher order” chloroplast origins. These are of course extensively dealt with in the main text.

Thank you for the comment. According to your suggestion, I have rewritten the last part of the Abstract as “Thus, there appear to be constraints on the evolution of those mechanisms, which likely began by incorporating photosynthetic cells before the establishment of chloroplasts by extending preexisting mechanisms to cope with oxidative stress originating from mitochondrial respiration and acquiring new mechanisms..” (L30–32).

(1-3) Line 76: “Mitochondria originated from alpha-proteobacterial endosymbiont that resided in a host cell...” Either: “Mitochondria originated from an alpha-proteobacterial endosymbiont that resided in a host cell...” or “Mitochondria originated from alpha-proteobacterial endosymbionts that resided in a host cell...”

I have fixed the error (L78).

(1-4) Line 156/157: “Whereas acquisition of photosynthetic organelle by primary endosymbioses are extremely rare in history, with only...” Same kind of issue: “Whereas acquisition of photosynthetic organelles by primary endosymbioses are extremely rare in history, with only...” And maybe change “rare in history” to “rare events”.

Thank you for the comment. I have changed “rare in history” to “rare events” (L165).

(1-5) Line 161: “...transform bacterial endosymbiont into chloroplasts...” Same kind of issue: “...transform bacterial endosymbionts into chloroplasts...”

I have fixed the error (L169).

Line 244-246: “Thus, upon acquisition of primary or complex chloroplasts, the eukaryotic hosts could have to some extent cope with ROS originating from photosynthesis by expanding or repurposing

existing systems.” Should read: “Thus, upon acquisition of primary or complex chloroplasts, the eukaryotic hosts could have coped to some extent with ROS originating from photosynthesis by expanding or repurposing existing systems.”

I have fixed the error (L253–254).

Line 318/319: “from non-photosynthetic eukaryotic ancestor and cyanobacterial ancestor of the primary chloroplast.” Should read: “from non-photosynthetic eukaryotic ancestors and cyanobacterial ancestors of the primary chloroplast.”

I have fixed the error (L326–327).

Line 324: “...much lower concentration of ascorbate...” Should read: “...much lower concentrations of ascorbate...” I’ll leave out further examples of this specific grammatical error.

I have fixed the error (L332) and checked and fixed others.

Line 699-701: “Furthermore, at least a few mechanisms for coping with photosynthetic oxidative stress have evolved even in organisms feeding on algae and that the associated genes may have been acquired by HGT from photosynthetic prey (Fig. 4; Table S1).” Should read: “Furthermore, at least a few mechanisms for coping with photosynthetic oxidative stress have evolved even in organisms feeding on algae and the associated genes may have been acquired by HGT from photosynthetic prey (Fig. 4; Table S1).”

I have fixed the error by removing the unnecessary “that” (L708).

Reviewer #2

(2-1) The author has made a thorough and conscientious effort to meet all of the reviewers’ suggestions. I think this is a fine, information-rich review. I mention two very minor additional suggestions:

Line 23: “Recent studies have shown that, besides algae and plant with innate chloroplasts,”: plants?

Line 38: It might be worth noting that Sanchez-Baracaldo et al. (2017, PNAS 114, E7737) suggest a much younger age for the origin of chloroplasts.

Thank you once again for taking your valuable time. I have addressed all the comments received as follows.

I have fixed the grammatical error (L23). In addition, I have cited the reference and added an explanation (L92–96).

I believe that the manuscript has been improved, thanks to the reviewers. I look forward to your feedback.

Yours sincerely,

Shin-ya Miyagishima